

**Regional and seasonal radiative forcing by perturbations to**
**aerosol and ozone precursor emissions**
**Nicolas Bellouin[1], Laura Baker[1], Øivind Hodnebrog[2], Dirk Olivié[3], Ribu**
**Cherian[4], Claire Macintosh[1], Bjørn Samset[2], Anna Esteve[1,*], Borgar Aamaas[2],**
**Johannes Quaas[4], and Gunnar Myhre[2].**
[1]{Department of Meteorology, University of Reading, Reading, United Kingdom}
[2]{Center for International Climate and Environmental Research – Oslo (CICERO), Oslo,
Norway}
[3]{Norwegian Meteorological Institute, Oslo, Norway}
[4]{Universität Leipzig, Leipzig, Germany}
[*]{now at: University of Valencia, Valencia, Spain}
Correspondence to: N. Bellouin (n.bellouin@reading.ac.uk)



## 1 Abstract

Dedicated model simulations by four general circulation and chemistry-transport models are
used to establish a matrix of specific radiative forcing, defined as the radiative forcing per unit
change in mass emitted, as a function of the near-term climate forcer emitted, its source
region, and the season of emission. Emissions of eight near-term climate forcers are reduced:
sulphur dioxide, the precursor to sulphate aerosols; black carbon aerosols; organic carbon
aerosols; ammonia, a precursor to nitrate aerosols; methane; and nitrogen oxides, carbon
monoxide, and volatile organic compounds, the precursors to ozone and to secondary organic
aerosols. The focus is on two source regions, Europe and East Asia, but the shipping sector
and global averages are also included. Emission reductions are applied over two time periods:
May-Oct and Nov-Apr. Models generally agree on the sign and ranking of specific radiative
forcing for different emitted species, but disagree quantitatively. Black carbon aerosols,
methane, and carbon monoxide exert positive specific radiative forcings. Black carbon exerts
the strongest specific radiative forcing, even after accounting for rapid adjustments from the
semi-direct effect, and is most efficient in local summer. However, although methane and
carbon monoxide are less efficient in a specific sense, the potential for decreasing the mass
emitted is larger. Organic carbon aerosols, sulphur dioxide, ammonia, and emissions by the
shipping sector exert negative specific radiative forcings, with local summer emission
changes being again more efficient. Ammonia is notable for its weak specific radiative
forcing. For aerosols, specific radiative forcing exerted by European emissions is stronger
than for East Asia, because the European baseline is less polluted. Radiative forcing of
European and East Asian emission reductions is mainly exerted in the mid-latitudes of the
Northern Hemisphere, but atmospheric transport yields sizeable radiative forcings in
neighbouring regions, such as the Arctic. Models disagree on the sign of the net radiative
forcing exerted by reductions in the emissions of nitrogen oxides and volatile organic
compounds, because those reductions trigger complex changes in the oxidising capacity of the
atmosphere, translating into radiative forcings by aerosols, methane, and ozone of different
signs. The response of nitrate aerosols to nitrogen oxide reductions is particularly important in
determining the sign of the corresponding radiative forcing. Model diversity comes from
different modelled lifetimes, different unperturbed baselines, and different numbers of species
and radiative forcing mechanisms represented. The strength of the aerosol-chemistry coupling
is also diverse, translating into aerosol responses to perturbations of ozone precursors of
different magnitudes.
Keywords: Radiative forcing; near-term climate forcers; Aerosols; Methane; Ozone
**1    Introduction**
Human activities have profoundly modified the composition of the atmosphere by increasing
the concentrations of long-lived greenhouse gases, such as carbon dioxide or
chlorofluorocarbons, and medium- to short-lived species, such as methane, tropospheric
ozone and aerosols. Once in the atmosphere, those species perturb the energy budget of the
Earth, exerting a radiative forcing (RF) of the climate system by various mechanisms which
depend on the species. Greenhouse gases, including methane, and ozone absorb longwave
radiation. Ozone is also an important absorber of shortwave radiation, mainly at ultraviolet
wavelengths. Aerosols scatter and absorb shortwave and, for larger particles, longwave
radiation: this mechanism is herein called aerosol-radiation interactions and denoted ari,
following Boucher et al. (2013). Changes in aerosol concentrations also translate in aerosol-
cloud interactions (aci) through changes in the number of cloud condensation nuclei,
modifying the radiative properties and life cycle of clouds. Aerosols that absorb shortwave
radiation, such as mineral dust and black carbon aerosols, also change the surface albedo
when depositing on snow or ice.
Figure 1 gives a simplified view of the tropospheric chemistry of ozone and its precursors. At
its heart are two equilibria: between nitrogen oxide (NO) and nitrous oxide ($NO_2$), and
between the hydroxyl (OH) and hydroperoxyl ($HO_2$) radicals. Photolysis of $NO_2$ is a source of
atomic oxygen which combines with molecular oxygen ($O_2$) to form ozone. Ozone, in turn,
undergoes photolysis to provide the principal source of OH. OH is converted into $HO_2$ during
oxidation of carbon monoxide (CO), $CH_4$, and other hydrocarbons. Then the reaction between
$HO_2$ and NO regenerates both $NO_2$ and OH. Nitric acid ($HNO_3$) and hydrogen peroxide
($H_2O_2$) are sinks of $NO/NO_2$ and $HO_2$, respectively. Tropospheric ozone is removed by dry
deposition. The lifetime of $CH_4$ is primarily determined by the abundance of OH, but volatile
organic compounds (VOCs) are also important. When the concentrations of $CH_4$, CO, and
VOCs increase, the limiting factor in the recycling of OH is the concentration of NO. VOCs



therefore control the rate of ozone production where $NO_X$ concentrations are not a limiting
factor. In addition, VOCs perturb the distribution of CH4 by modifying OH concentrations.
Finally, OH, $O_3$, $HO_2$, and $H_2O_2$ are also involved in the dry and in-cloud oxidation of $SO_2$,
the gaseous precursor to sulphate aerosol. Those tight interactions between species add
components to the RF caused by feedbacks of one species onto another: for example, changes
in methane concentrations trigger changes in tropospheric ozone, which exerts its own RF,
called primary-mode ozone RF (Prather, 1996). Shindell et al. (2009) found sizeable impacts
of nitrogen oxides ($NO_X$), CO, and $CH_4$ emissions on aerosol formation in global simulations
of atmospheric chemistry with the Goddart Institute for Space Studies (GISS) model. The
crucial link in those chemical feedbacks is OH, which in spite of its very short lifetime plays
an important role in both the atmospheric chemistry of ozone and the oxidation of aerosol
gaseous precursors.
The fifth assessment report of the Intergovernmental Panel on Climate Change (IPCC)
formalised two RF concepts: the stratospherically-adjusted and effective RF (Boucher et al.,
2013; Myhre et al., 2013a). In the first definition, surface and tropospheric conditions are held
fixed to their unperturbed state, but stratospheric temperatures are allowed to adjust. In
contrast, effective RF (ERF) also includes rapid adjustments to the tropospheric state. Those
rapid adjustments occur on shorter timescales than deep ocean and sea ice changes and
include such processes as the change in cloud cover that follows the local atmospheric
warming caused by aerosol absorption of shortwave radiation, the change in cloud cover due
to aerosol-driven changes in precipitation efficiency, the increased spring melting that follows
black carbon deposition on snow, or the change in cloud cover that immediately follows
changes in thermodynamic profiles in response to an increase in carbon dioxide
concentrations. Because ERF includes rapid adjustments, it is a better indicator of the
eventual surface temperature response than RF. It has however recently been suggested that
an additional efficacy, acting not on the RF but on temperature change itself, is needed to
account for differences in global patterns of ERF, whereby forcing agents located in the
Northern Hemisphere cause more rapid land surface temperature responses (Shindell, 2014).
Both stratospherically-adjusted and effective RF exclude the radiative impact of large-scale
changes in sea surface temperatures, which are part of the climate response.



Climate metrics such as Global Warming Potential and Global Temperature Change Potential
(e.g. Shine et al., 2005) have been developed to compare the climate impact of the emissions
of different species, trying to account for both the radiative forcing efficiency of the species
and their lifetime. Emission-based climate metrics allow the exploration of future mitigation
options where the basket of species emitted by a given sector of activity changes in response
to policies and technological advances. Those metrics require the knowledge of
stratospherically-adjusted RF exerted per change in unit mass emission rate, hereafter called
specific RF (SRF) and given in mW m$^{-2}$ (Tg yr$^{-1}$)$^{-1}$.
In the past, the available literature has been used in a rather ad-hoc way to quantify SRF.
Table 1 summarises estimates from five previous multi-model studies. Bond et al. (2013)
assessed black carbon (BC) emissions and RF over the period 1750—2005 based on multi-
model estimates scaled to remove biases in absorption aerosol optical depth against present-
day observations. Myhre et al. (2013b) conducted a multi-model inter-comparison of RFari
over the period 1850—2000 by the second phase of the AeroCom (Aerosol Comparisons
between Observations and Models) initiative. Shindell et al. (2013) and Stevenson
et al. (2013) document multi-model inter-comparisons of aerosol and ozone RF, respectively,
over the period 1850—2000 by the Atmospheric Chemistry and Climate Model Inter-
comparison Project (ACCMIP, Lamarque et al., 2013). Finally, Yu et al. (2013) report results
of a multi-model inter-comparison of RF due to 20% reductions in the emissions of 4 regions
as part of the Hemispheric Transport of Air Pollution (HTAP) project. Table 1 also shows
results for the present study to allow for an easy comparison: results are discussed in Sect. 4.
All studies agree on the sign of the SRF of individual species. All near-term climate forcers
(NTCFs) exert positive SRFs, which lead to a gain in energy for the climate system when
emissions are increased, except for sulphate, organic carbon (OC), and nitrate aerosols, and
nitrogen oxide gases, which exert negative SRFs. According to those studies, BC exerts the
strongest SRF of all NTCFs, in absolute values. Compared to other aerosol species, its SRF is
an order of magnitude larger. Among ozone precursors, the SRF of nitrogen oxides (NO$_X$) is
the strongest, being for example about 16 times larger than and of opposite sign to CO SRF. It
is important to note that the strength of the SRF of a given NTCF is only one aspect of its
climate impact: the strength of anthropogenic emission rates is also important. Therefore, the
strong SRFs of BC and NO$_X$ have to be considered in the context of their weak emission rates





relative to other NTCF precursors like sulphur dioxide ($SO_2$) and CO. SRF are however
useful to characterise the forcing characteristics of individual species in a context where the
amount of species emitted by a given economic sector varies in time because of changes in
technology or legislation (e.g. Smith et al., 2013).
The SRF estimates by Yu et al. (2013) are given for four regions (North America, Europe,
East Asia, South Asia), in contrast to the other studies listed in Table 1, which only take a
global view. They identified regional differences that appeared robust across participating
models. East Asian $SO_2$ emissions exert an SRF that is only 75% of that by European
emissions, a smaller value that Yu et al. (2013) attribute to a limitation in sulphur-cycle
oxidants over East Asia, which suppresses conversion of $SO_2$ to sulphate aerosols in that
region. Furthermore, their estimate of BC SRF from European emissions is 30% stronger than
that of other regions, a result attributed to the geographical extent of European aerosol
transport, which covers in particular the bright surfaces of the Arctic and Sahara, where BC
aerosols exert a strong positive RF.
Diversity in SRF estimates reflect diverse choices made when representing tropospheric
aerosols and chemistry in global models, uncertainties in understanding and representations of
radiative forcing mechanisms, and impacts of differences in other aspects of the host models.
Myhre et al. (2013b) found a sizeable range in aerosol optical properties among AeroCom
models. Also, different simulated vertical profiles affect RF efficiency, especially for
absorbing aerosols. For BC, Samset et al. (2013) suggest that different vertical distributions
explain 20% of RF diversity among AeroCom models. More generally for RFari interactions,
Stier et al. (2013) attribute a third of total AeroCom diversity to host model differences,
mainly coming from differences in cloud distributions, surface properties, and radiative
transfer, the latter contributing up to 20% (Randles et al., 2013). For RF of aerosol-cloud
interactions (RFaci), additional diversity comes from the parameterised sensitivity of cloud
albedo to aerosol changes, which varies by more than a factor 4 in AeroCom models (Quaas
et al., 2009). For ozone RF, Stevenson et al. (2013) estimate an overall uncertainty of 30% by
accounting for uncertainties in radiative transfer calculations (radiation schemes, simulated
clouds, and stratospheric adjustment techniques), diversity in simulated ozone levels in
present-day and pre-industrial conditions, uncertainties due to the identification of the





tropopause level, and uncertain climate change impacts on atmospheric dynamics and
chemistry.
Diversity is also introduced by the different treatment of modelled estimates in different
studies. For example, Myhre et al. (2013b) include all participating models in their estimates,
but Shindell et al. (2013) only select those models able to satisfactorily represent present-day
aerosol distributions and recent trends. Bond et al. (2013) scale modelled RF towards stronger
values mainly through increases in emissions to account for a perceived low bias in simulated
BC concentrations and absorption aerosol optical depth. This upward scaling has been
challenged by recent studies (Wang et al., 2014a; Samset et al., 2014), which argue that a
suite of observational constraints, including vertical profiles of BC concentrations in remote
regions, does not support the stronger end of the range in BC RF. Wang et al. (2014a) reduce
substantially the BC underestimation in their model compared to observations by improving
the model horizontal resolution. Samset et al. (2014) suggest that a majority of the aerosol
models overestimate BC lifetime, causing too strong an RF.
Conversely, multi-model inter-comparison studies do not generally consider all sources of
diversity in SRF estimates. Those studies typically mandate a common set of anthropogenic
emissions for all participating models, for example the emission dataset of Lamarque et al.
(2010) is used by AeroCom 2 and ACCMIP simulations. However, both present-day and pre-
industrial emissions are uncertain and that uncertainty, which can be substantial (Carslaw et
al., 2013), is not reflected in many of the ranges given in Table 1. Yu et al. (2013) include the
effect of different emission datasets being used by different models, but exclude another
source of diversity by using a single set of aerosol optical properties to derive their RF
estimates. The variable experimental designs of the studies listed in Table 1 hinder a clean
assessment of the metrics uncertainty caused by diversity in RF estimates (Fuglestvedt et al.,

25     2010).

The Evaluating the CLimate and Air Quality ImPacts of Short-livEd Pollutants (ECLIPSE)
project (Stohl et al., 2015) covered in a consistent way the causal chain that links emissions of
NTCFs to their SRFs (this study), climate metrics (Aamaas et al., 2015) and the subsequent
climate response (Baker et al., 2015). This study documents the calculation of the ECLIPSE
matrix of SRF for NTCFs. Its aim is to provide a dataset of SRFs that spans the diversity



among models, while providing an explanation, albeit sometimes incomplete, for that
diversity.
SRFs are calculated for reductions in the anthropogenic emissions of primary aerosols (BC,
OC), aerosol precursors (sulphur dioxide, ammonia), ozone and secondary aerosol precursors
(NO$_X$, CO, VOC), and methane. This study shares common aspects with those listed in
Table 1, for example in the range of NTCFs considered. It is also a multi-model study, with
4 global circulation or chemistry-transport models participating. Like Yu et al. (2013), it takes
a regional view by focusing on two source regions, Europe and East Asia, and singling out the
shipping sector. But other aspects of this study are more distinctive. Emissions are perturbed
seasonally, to assess which of local summer or wintertime emission reductions are most
effective at exerting an SRF. This study also quantifies a larger number of radiative
mechanisms: RFaci is systematically included, and ozone precursor RFs include a
contribution from aerosol changes that arise through aerosol-chemistry couplings.
Contributions to BC RF from deposition on snow and rapid adjustments from the semi-direct
effect are also estimated, albeit from a single model. Finally, although uncertainties in
emissions are not fully accounted for because all models participating in this study share the
same emission datasets, the large uncertainties in pre-industrial emissions are irrelevant here,
because the perturbations used in this study are defined as percentages of present-day
emissions.
The paper is structured as follows. Section 2 describes the participating models and
experimental design. Section 3 quantifies the components of SRF simulated by each model as
a function of emitted species, region, and season. Causes of model diversity are also identified
and discussed. Section 4 gives the best estimate of the SRF matrix resulting from the
ECLIPSE project. Finally, Sect. 5 concludes with a discussion of research priorities for
decreasing model diversity, and the challenges encountered when quantifying rapid
adjustments. Supplementary Figures show annually-averaged distributions of RF components
for all perturbations.



## 2    Models and experimental protocol

Participating models are ECHAM6-HAM2, HadGEM3-GLOMAP, NorESM1, and OsloCTM2. It is known from previous participations of those models in multi-model inter-comparisons, including the studies cited in the introduction, that the four models span a large range of inter-model diversity for both aerosol and ozone. Horizontal and vertical resolution, and the number of aerosol species included depend on the model (Table 2). In particular, OsloCTM2 is the only model that represents nitrate aerosols. ECHAM6 does not simulate secondary organic aerosols, and also lacks interactive ozone chemistry, and thus did not perform perturbations to ozone precursor emissions.

ECHAM6-HAM2 is the European Centre for Medium-Range Weather Forecasts Hamburg model version 6 (Stevens et al., 2013). Its radiation scheme is RRTM-G (Iacono et al., 2008). Aerosols are represented by the two-moment Hamburg Aerosol Model (HAM) version 2 (Zhang et al., 2012), which consists of the microphysical module M7 that simulates seven internally-mixed aerosol modes (Vignati et al., 2004; Stier et al., 2005). Aerosol interactions with liquid and frozen water clouds follow Lohmann et al. (2007).

HadGEM3 is the Hadley Centre Global Environment Model version 3 (Hewitt et al., 2011). Its radiation scheme is described by Edwards and Slingo (1996). Gas-phase chemistry is modelled by the United Kingdom Chemistry and Aerosols (UKCA) TropIsop scheme, which treats 55 chemical species (37 of which being transported) including hydrocarbons and isoprene and its degradation products (O'Connor et al., 2014). Aerosols are coupled to the chemistry, and modelled by UKCA-GLOMAP (GLobal Model of Aerosol Processes, Mann et al., 2010), which represents the size-resolved internal mixture using a two-moment modal approach and four soluble and insoluble aerosol modes. Aerosols interact with liquid clouds only, following the empirical relationship between aerosol number and cloud droplet number concentration established by Jones et al. (1994).

NorESM1-M is the Norwegian Earth System Model version 1 (Bentsen et al., 2013; Iversen et al., 2013). Its atmosphere and aerosol module is CAM4-Oslo (Kirkevåg et al., 2013) and the radiation scheme is described by Collins (2001). In the version used in this study, aerosols (described by 20 tracers) are fully coupled to the MOZART tropospheric gas-phase chemistry scheme (Emmons et al., 2010), which treats 84 gaseous species. Aerosol mass concentrations are simulated in four size classes: nucleation, Aitken, accumulation, and coarse modes.



OsloCTM2 is the CTM of the University of Oslo and the Center for International Climate and
Environmental Research – Oslo (CICERO) (Myhre et al., 2009; Skeie et al., 2011). The
model is driven by meteorological data generated by the Integrated Forecast System (IFS)
model at the European Centre for Medium-Range Weather Forecasts (ECMWF). The model
simulates the tropospheric chemistry of 67 species (Dalsøren et al., 2007). Aerosols are
simulated as external mixtures of 7 aerosol types, including nitrate, as described by Skeie et
al. (2011). RFari and RFaci are computed by offline radiative transfer calculations, as
described in Myhre et al. (2007) and Skeie et al. (2011). Myhre et al. (2000) describes the
offline calculations performed to obtain ozone radiative forcing.
The 48 ECLIPSE RF simulations are listed in Table 3. Simulations are only 1-year long (after
spin-up) because RF by definition excludes changes in the tropospheric state and inter-annual
differences in meteorology are the only source of variability between simulations.
Meteorology affects transport and removal processes, especially wet deposition, and to a
lesser extent chemical production when driven by temperature or availability of sunlight.
Perturbation simulations made with HadGEM3 were extended to 3 years and suggest that
inter-annual variability never exceeds ±10 % of globally-averaged RF, which is small
compared to inter-model diversity. RF is calculated at the top of the atmosphere as the
difference in net shortwave and longwave radiative fluxes between the perturbed and control
simulations.
Control emissions are taken from the ECLIPSE dataset version 4a (Stohl et al., 2015) for the
year 2008. A seasonal cycle has been applied to the emissions of the domestic sector, to
reflect changes in heating as a function of temperature. This seasonal cycle is obtained by
multiplying annual total domestic sector emissions by a gridded dataset of monthly weights,
obtained by the Mitigation of Arctic warming by Controlling European Black carbon
emissions (MACEB) project following Sect. 3.3 of Streets et al. (2003), where stove
operation times are expressed as a function of climatological monthly-mean temperature.
Emission perturbations involve a 20% decrease of primary and precursor emissions of the
given species in one of the following regions: Europe, East Asia, shipping, and Rest of the
World (RotW). Results for that last region are not presented directly in this paper: instead,
global results are given by adding Europe, East Asia, and RotW together. Applying a
decrease, rather than an increase, has been chosen because it better represents air quality and





climate policy objectives. The value of 20% was chosen to be representative of typical
technologically feasible emission reductions. The same value was also used in previous
HTAP simulations (Yu et al., 2013). The definition of regions also follows HTAP, more
specifically tier-1 HTAP regions. Regions are shown in Figure 2. Here, Europe includes
European Union and European Economic Area countries, and Switzerland, Turkey, and
former Yugoslavia. East Asia includes China, Japan, Taiwan, North and South Korea, and
Mongolia. Because of the specificity of the shipping sector in the policy agenda, its emissions
have been perturbed independently, with all species emitted by that sector being perturbed
together, although OsloCTM2 and NorESM1 have run perturbations for each species within
the shipping sector (results not shown). Shipping emissions are taken from the RCP6.0 dataset
(Fujino et al., 2006) prepared for phase 5 of the Climate Model Inter-comparison project
(CMIP5), interpolated to 2008 between 2005 and 2010. All perturbations are applied either in
Northern Hemisphere summer (May-October) or winter (November-April). The size of the
emission perturbations is given in Table 3, except for the shipping sector perturbations, which
are shown in Table 4. The size of shipping emission perturbations is different for ECHAM6-
HAM, because RCP8.5 (Riahi et al., 2007) was used, and for $NO_X$ in NorESM1, because of a
mistake when processing that particular dataset. The size of non-methane VOC emission
perturbations is model-dependent because the list of species emitted under the VOC label
depends on the model used: 5 for HadGEM3, 14 for NorESM1, and 12 for OsloCTM2. For
OsloCTM2, VOC emissions were converted to unit mass of carbon by assuming a mean VOC
atomic weight of 47 u.
Methane perturbations are achieved by scaling the prescribed concentrations or mass-mixing
ratios, rather than by perturbing emissions like for the other NTCFs. This difference in
treatment arises because HadGEM3, NorESM1, and OsloCTM2 prescribe methane
concentrations at the surface and then let the chemistry scheme determine the vertical
distribution, thus avoiding long spin-ups caused by the 12-year lifetime of methane in the
atmosphere. Scaled methane surface concentrations $C$ are given by the equation:

28           $C = C_0 \cdot (E/E_0)^f$                                        (Eq. 1)

where $C_0$ are the control surface concentrations, $E$ is the global emission rate where the
anthropogenic contribution has been reduced by 20%, and $E_0$ is the control global emission
rate. $E/E_0$ is therefore equal to 0.8 in this study. $f$ is the feedback factor of methane on its own



lifetime, defined as the ratio of methane perturbation lifetime to total budget lifetime. The
value of $f$ for each participating model was not known when preparing the simulations, and
was therefore taken at 1.34 following Holmes *et al.* (2013). As discussed in Sect. 3.5, actual
values of $f$ range from 1.28 to 1.46, in reasonable agreement with the value initially assumed.
Because the long atmospheric lifetime of methane allows it to be well mixed geographically,
methane perturbations are not applied regionally. NorESM1 applied perturbations seasonally
(May—Oct and Nov—Apr) and found differences in SRF of only 7% between the two
seasons. Because that seasonal dependence is small, OsloCTM2 and HadGEM3 have applied
the perturbation for the whole year.
Three methods are used to obtain stratospherically-adjusted RF from the perturbation
simulations, depending on the species being considered and whether the model is capable of
interactive radiation calculations (Table 1).
• To obtain the RF of aerosol perturbations in general circulation models, the model
evolution ("meteorology") is set to be independent of the perturbation. The method used
to achieve this independence involves diagnosing radiative fluxes with and without the
forcing agent included, with the second set of radiative fluxes used to advance the model
into its next time step. Stratospheric adjustment is neglected for aerosols, because
tropospheric aerosol perturbations have little effect on stratospheric temperatures.
Aerosol RF includes both ari and aci, except for ECHAM6, which only diagnosed ari.
• To obtain the RF of aerosol perturbations in chemistry-transport models and the RF of
ozone exerted by ozone-precursor perturbations in all models, instantaneous RF is
computed by offline radiative transfer codes, using aerosol and trace gas distributions
obtained from the perturbation simulations. HadGEM3 ozone RF is computed with the
offline version of the radiative transfer code by Edwards and Slingo (1996). OsloCTM2
aerosol and ozone RF, and NorESM1 ozone RF, are computed with offline longwave and
shortwave radiative transfer codes as described earlier in this section. For all models,
ozone RF is adjusted for changes in stratospheric temperatures.
• The RF of methane is computed using the analytical expression established by Myhre et
al. (1998), which accounts for stratospheric adjustments. Details of this calculation are
given in Sect. 3.5 below.



The four models simulate different aerosol and tropospheric ozone lifetimes, as shown in
Table 5. Sulphate aerosol lifetime varies by a factor 1.5, from 3.5 to 5.2 days. All models
agree that BC lifetime is longer, with a diversity similar to sulphate, with variations by a
factor 1.5, from 5.2 to 8.0 days. Modelled OC lifetime is also longer than that of sulphate,
with a large diversity, with variations by a factor 2.5, from 3.1 to 7.7 days. Tropospheric
ozone lifetime is also diverse: HadGEM3 and NorESM1 disagree by a factor 1.3, from 20.7 to
26.4 days. OsloCTM2 did not diagnose it. Differences in simulated lifetimes are thought to
arise from virtually all aspects of the models, including differences in the simulated present-
day climate, the treatment of atmospheric horizontal and vertical transport, atmospheric
chemistry, and wet and dry deposition processes. Large model spreads have long been a
characteristics of aerosol and chemistry inter-comparisons (e.g. Myhre et al., 2013b;
Stevenson et al., 2013), in part because of a lack of strong observational constraints on
atmospheric lifetimes on a global scale (Kristiansen et al., 2012; Hodnebrog et al., 2014). The
four ECLIPSE models are representative of those spreads, which will affect the range of SRF
they simulate.

## 2.1   Biases and scaling of specific radiative forcing

Aerosol and ozone distributions simulated by the four models participating in this study have
been compared to observations as part of their development cycles, multi-model inter-
comparisons, and within the ECLIPSE project. Those evaluations draw a complex picture,
where model skill at reproducing NTCF distributions with fidelity differs among models and
is strongly region- and species-dependent.
To ensure the relevance of evaluations to the SRF matrix, they are discussed for each species
alongside SRF results in Sect. 3. This section focuses on the evaluation of two cross-cutting
quantities: aerosol optical depth (AOD) and ozone concentrations. To reproduce the
magnitude and geographical distribution of those two quantities accurately, one needs to
simulate several aerosol species and ozone precursors well.
ECLIPSE evaluation of AOD focused on East Asia. Patterns of total AOD simulated by the
four models over East Asia compare well against MODIS (Moderate Resolution Imaging
Spectroradiometer) aerosol retrievals, but fail quantitatively to various extents depending on





model, season, and region (Quennehen et al., 2015). NorESM1 and, to a larger extent,
HadGEM3 overestimate the AOD background. HadGEM3 also overestimates AODs over
East Asia, while they are underestimated by ECHAM6 and OsloCTM2. Models are similarly
diverse in their ability to reproduce the temporal variability of AOD: NorESM1 ranks best in
Eastern China, but HadGEM3 and OsloCTM2 do better in northern India. In the vertical,
comparing the four models to the vertical profiles of aerosol scattering retrieved by the Cloud-
Aerosol Lidar with Orthogonal Polarization (CALIOP) shows that models underestimate
aerosol scattering in the lowest 2 km of the atmosphere, but overestimate it at altitudes
ranging from 2 to 4 km, hinting at too efficient a transport into the free troposphere or too
weak sinks (Quennehen et al., 2015). For RFaci, such errors in simulated vertical profiles may
lead to too weak an SRF, because aerosols that should have interacted with clouds end up
being simulated above those clouds, unable to interact with them. For BC, placing the
aerosols too high in the atmosphere leads to overestimating RFari (Samset et al., 2013) but
underestimating rapid adjustments from semi-direct effects, so the net impact on SRF depends
on the local balance between those two mechanisms.
For tropospheric ozone, Schulz et al. (2015) evaluated the surface concentrations of ozone
simulated by OsloCTM2 and NorESM1 models over Europe. They find that OsloCTM2
reproduces magnitude and seasonality very well, but that NorESM1 has a high bias,
especially in Northern Hemisphere winter. HadGEM3 could not be part of that evaluation, but
an evaluation of an earlier version of the model indicates that it captures well the seasonality
and magnitude of ozone surface concentrations at mid-latitude of the NH, but has a high bias
in the Tropics and high latitudes (O'Connor et al., 2014). Over Asia, Quennehen et al. (2015)
find that HadGEM3, NorESM1 and OsloCTM2 simulate total column ozone within 10%,
with OsloCTM2 having essentially no bias against infrared sounder observations and
HadGEM and NorESM1 both overestimating the ozone column. The three models locate
ozone too low in the troposphere, as evidenced by an increase in positive biases by 3 to 7% in
the 0—6 km layer. The models are able to qualitatively reproduce the gradients existing
between surface concentrations in urban and rural conditions, but modelled gradients are
smoothed out because of the relatively coarse resolutions of the models.
RF and SRF cannot be evaluated against observations, so the challenge is to interpret what
regional evaluations of surface concentrations, vertical profiles and optical properties imply



for globally-averaged SRF to regional perturbations. As a general guideline, biases in mass
concentrations or optical depth should not be expected to propagate fully into biases in SRF,
because of its normalisation to emission rates. Biases would not propagate in regions where
concentrations and AOD scale linearly with emissions, for example where oxidants do not
limit sulphate or ozone formation, and where RF scales linearly with concentrations and
AOD, as is for example mostly the case for aerosol-radiation interactions. However, biases in
lifetime and local concentrations become important on a global scale when a model fails to
transport an NTCF to a region where RF efficiency is significantly different to its source
region, for example where surfaces are highly reflective (deserts, ice, and snow) or the cloud
regime is strongly susceptible to aerosol influences (low maritime clouds).
In addition, there are known non-linearities in the emission-concentration-RF chain. Methane
RF is proportional to the square root of its concentration (Myhre et al., 1998). Aerosol-cloud
interactions are non-linear because strong non-linearities between aerosol and CCN
concentrations (Hegg, 1994) combine with strong non-linearities between cloud droplet
concentrations and cloud albedo (Taylor and McHaffie, 1994). Aerosol-cloud interactions will
therefore desaturate more quickly in regions where aerosol concentrations are small to
moderate: this is where biases in concentrations will have strong impacts on RF estimates.
Conversely, errors in simulating the large concentrations in polluted regions will have a lesser
impact. In constrast, errors in simulating vertical profiles can have disproportionate impacts
on SRF for species that absorb in the shortwave spectrum. Locating BC aerosols too high up
in the atmosphere will systematically overestimate their RF because aerosol absorption is
enhanced when overlying bright surfaces such as clouds. Hodnebrog et al. (2014) show that
reducing lifetime by 40% to 3.9 days to improve the match to observed BC vertical profiles
reduces the forcing approximately by a factor of two. Ozone adjusted RF is similarly altitude-
dependent because of varying degrees of compensation between the shortwave and longwave
components of RF. In the troposphere, where this study's ozone perturbations are located,
ozone RF effiency increases with altitude with a maximum near the tropopause (Lacis et al.,
1990). The fact that the models locate ozone too low in the atmosphere would therefore
introduce a low bias in the SRF exerted by ozone precursors.
To summarise, regional and seasonal variations in model skill at simulating aerosols and
ozone with fidelity, the normalised nature of SRF, and non-linearities in the emission-to-





forcing chain preclude a simple scaling of modelled SRF with identified biases. So this study
reports SRF as simulated by the models, discussing where appropriate in the next section the
implications of comparisons to observations for the SRF exerted by each species.

## 3   Specific radiative forcing by species

In this section, SRFs of each primary or precursor species are discussed in turn. SRF is
stratospherically-adjusted but excludes rapid adjustments in the troposphere, with one
exception: rapid adjustments of BC semi-direct effects have been computed independently
and are discussed in Sect. 3.2. SRF is given for May-October (hereafter labelled Summer for
the sake of simplicity but also because emission perturbations are disproportionality located in
the Northern hemisphere) and November-April (labelled Winter), for three regions (Europe,
East Asia, and Global), and for the shipping sector. Globally-averaged RF is computed as the
sum of the European, East Asian, and RotW perturbations. Although perturbations are not
exactly additive, this is a good first-order assumption.

## 3.1   Sulphur dioxide

Figure 3 shows globally- and annually-averaged SRF for sulphur dioxide perturbations in the
four ECLIPSE models. $SO_2$ SRF ranges from $-1.2$ mW m$^{-2}$ (Tg[$SO_2$] yr$^{-1}$)$^{-1}$ for the European
Winter perturbation by ECHAM6 to $-18.0$ mW m$^{-2}$ (Tg[$SO_2$] yr$^{-1}$)$^{-1}$ for the European Summer
perturbation by HadGEM3. ECHAM6 is consistently associated with weak SRF because it
only diagnoses ari. On a global scale, ari account for 43% and 55% of total RF according to
NorESM1 and OsloCTM2, respectively, which diagnosed ari and aci separately. The ari
contribution varies seasonally, being generally stronger in Winter than in Summer
perturbations in both models. They also vary regionally, but in a model-dependent way:
OsloCTM2 obtains the largest contribution of ari in Europe, but NorESM1 locates it in East
Asia. Note that only ari and aci RF mechanisms are considered here: ozone and methane RF
exerted by perturbations to OH distributions resulting from perturbations of $SO_2$ emissions are
negligible.
All models agree that $SO_2$ SRF is stronger for Summer than Winter perturbations, which is
expected because sulphate RFari and RFaci act almost exclusively in the shortwave spectrum



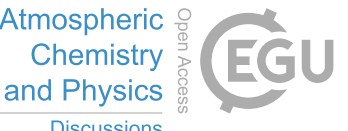

and are therefore a strong function of solar irradiance. As shown in Figure S1, sulphate RF
covers a larger area in models with longer sulphate aerosol lifetimes, such as HadGEM3, than
in models with shorter lifetimes, like OsloCTM2. This extended coverage has two competing
effects on the strength of SRF, both driven by non-linearities in RFaci. On the one hand, a
longer lifetime promotes stronger RFaci because emission perturbations propagate more
easily to remote regions where concentrations are low and RFaci desaturates more easily. On
the other hand, a longer lifetime weakens RFaci by increasing concentrations in the reference
simulation in those same remote regions, saturating RFaci. The first effect dominates in the
ECLIPSE models because SRF strength increases with lifetime. RFaci non-linearities also
explain why models simulate weaker SRFs for East Asian than European perturbations. With
a more polluted baseline, East Asian aci stands more often at the saturated end of the CCN-
cloud albedo relationship, where RFaci is weak (Wilcox et al., 2015).
Non-lifetime related aspects will also increase diversity in $SO_2$ SRF. Firstly, the
parameterised strength of aci varies between models (Quaas et al., 2009). Secondly, models
have different cloud climatologies (Jiang et al., 2012): aerosols being more likely to be co-
located with low clouds, low cloud amounts influence RFaci. Thirdly, the relatively weak $SO_2$
SRFs simulated by OsloCTM2 are partly due to the inclusion of nitrate aerosols in that model.
Ammonium nitrate formation is suppressed by that of ammonium sulphate, meaning that a
reduction in $SO_2$ emissions indirectly favours nitrate formation (Bellouin et al., 2011). The
RF exerted by the increase in nitrate aerosol weakens the overall SRF. In OsloCTM2, nitrate
RFari offsets 4 to 10% of sulphate RFari, with larger offsets obtained in Northern Hemisphere
winter months.
In Europe, ECLIPSE models underestimate sulphate aerosol surface concentrations, in part
because of underestimated $SO_2$ oxidation rates (Schulz et al., 2015). In the Arctic, Eckhardt et
al. (2015) report that models underestimate winter/spring sulphate concentrations by an
average factor of 2, with longer-lived aerosols yielding no improvement against observations,
suggesting that transport of aerosols to the Arctic is only part of the problem. Although those
concentration biases may not affect SRF, the fact that models fail to reproduce the observed
summer minimum in the Arctic, as reported by Eckhardt et al. (2015), suggests that SRF may
be too weak in magnitude in that region.





Changes in sulphate aerosols may exert rapid adjustments that follow the perturbation to
cloud droplet size distributions. Those are not quantified here, because ECLIPSE models do
not yet adequately represent the observed dependence of the strength and sign of rapid
adjustments on cloud regime (Christensen and Stephens, 2011). In addition, rapid adjustments
are difficult to isolate robustly from internal variability in top-of-atmosphere radiative fluxes,
especially for the small perturbations imposed in this study.
## 3.2   Black carbon aerosol
Figure 4 shows globally- and annually-averaged SRF for BC aerosol perturbations in the four
ECLIPSE models. BC SRF ranges from +3.7 mW m$^{-2}$ (Tg[C] yr$^{-1}$)$^{-1}$ for the East Asian Winter
perturbation by HadGEM3 to +139.8 mW m$^{-2}$ (Tg[C] yr$^{-1}$)$^{-1}$ for the Global Summer
perturbation by the same model. BC SRF is positive because ari and snow-albedo
mechanisms both exert positive RFs due to the strong absorption of shortwave radiation
associated with BC aerosols. However, negative RF from aci and rapid adjustments from the
semi-direct effect partly offset those positive contributions. Note that changes in primary BC
emissions are also associated with ozone and methane RF because of changes in the surface
of aerosols available for heterogeneous reactions, but the resulting RFs are negligible and not
reported in this study. The relatively weak BC SRF by ECHAM6 is partly due to the lack of
diagnosis of RFaci, although ari strongly dominate, with RFaci representing only 2 to 15% of
its RFari according to NorESM1 and OsloCTM2.
BC SRF is stronger for Summer than Winter perturbations in all models, which is expected
for RF mechanisms that act primarily on shortwave radiation. BC SRF is not clearly
correlated to modelled lifetimes, contrary to what is found for sulphate. The lack of
correlation has three main causes. First, the strength of BC RFari is strongly modulated by the
albedo of the underlying surface: for a given optical depth and single-scattering albedo, BC
aerosols exert a stronger RFari when located above bright surfaces, like deserts, snow and ice,
and clouds. So BC transport to remote regions by longer lifetimes affects the diversity of
surface albedos experienced by the BC aerosols in a complex way. Figure S2 shows for
example that the long lifetime of BC in NorESM1 translates into a strong RF over the Arctic
for East Asian and Global perturbations. Other models have weaker RFs because the same





perturbations affect the Arctic in a weaker way. NorESM1's BC lifetime may however be too
long, because it overestimates BC concentrations in the Arctic (Eckhardt et al., 2015).
Secondly, BC mass-absorption coefficients (MAC) vary among models because those models
make different prescriptions of refractive indices and different assumptions of mixing state
and hygroscopic growth. Globally-averaged BC MAC for ambient conditions is 10.4 $m^2\,g^{-1}$ in
ECHAM6, 15.7 $m^2\,g^{-1}$ in HadGEM3, only 3.8 $m^2\,g^{-1}$ in NorESM1, and varies between
7.3 $m^2\,g^{-1}$ for hydrophobic and 11.0 $m^2\,g^{-1}$ for hydrophilic BC in OsloCTM2. Finally, the
impact of the aerosol mixture can be complex. For example, HadGEM3 simulates negative
BC SRFs over northern Russia for the Global Summer perturbation (Figure S2). The reason is
that perturbations of primary BC emissions also perturb condensation of organic materials in
that model: by having fewer primary particles, the gaseous condensation sink is suppressed,
favouring the nucleation of new CCNs in pristine regions (Bellouin et al., 2013).
ECLIPSE models generally underestimate BC aerosol surface concentrations in Europe
(Schulz et al., 2015). In the Arctic, Eckhardt et al. (2015) find underestimates by factors 4 to
27 in the four models, especially HadGEM3, in all seasons, except for NorESM1 which
overestimates concentrations by a factor 1.3 in summer. Those incorrect estimates occur
throughout the troposphere, although aircraft campaigns are biased high by targetting
biomass-burning aerosol plumes rather than average background conditions. Eckhardt et al.
(2015) conjecture that emissions are underestimated, especially in northern Russia. Modelled
BC has not been evaluated in India, but Gadhavi et al. (2015) also find that the BC emission
rates used in ECLIPSE are likely underestimated. BC being a primary aerosol, it is therefore
probable that Indian concentrations are underestimated as well. Those biases would not bias
the SRF unless radiative efficiency is also biased, and there are indications that models
systematically overestimate BC concentrations in the remote troposphere, especially at higher
altitudes (Samset et al., 2014). The SRF would therefore be too strong. Hodnebrog et al.
(2014) confirm this assessment and argue that BC emissions are underestimated in models
while lifetimes are too long. They find that those overestimated lifetimes lead to an
overestimate of BC SRF by a factor of around 2.
Absorbing AODs offer an indirect constraint on BC concentrations, at least in regions where
mineral dust aerosols are not present and assuming that OC aerosols do not strongly
contribute to absorption, which may not be true in biomass-burning regions (Saleh et al.,



2014). Compared to AERONET retrievals, the four ECLIPSE models underestimate
absorbing AOD by more than a factor 2 (Schulz et al., 2015). The constraint brought by
AERONET absorption retrievals is however debated because its retrieval algorithm requires
large solar zenith angles and has large uncertainties at low AODs (Dubovik et al., 2000).
AERONET absorbing AODs are therefore only reported for morning/evening conditions and
for thicker plumes, which may introduce systematic biases by not sampling background
aerosols. In addition, Wang et al. (2015) showed that the fairly low resolutions of global
models similar to those used in this study induce an artificial negative bias when comparing to
AERONET stations in Asia.
The RF due to BC deposition on snow, shown in grey in Figure 4, is only quantified by
OsloCTM2. It is a small term globally, and negligible for Summer perturbations because
fossil-fuel BC sources are mostly located in the Northern Hemisphere where snow cover is
minimum during May to October. Winter perturbations exert stronger BC-on-snow RFs,
which represent 15, 20, and 53% of RFari for Global, East Asian, and European perturbations,
respectively. The disproportionately strong contribution of the European perturbation is due to
its geographical location: in spite of smaller BC emitted mass in Europe, Arctic RF is similar
to that of East Asian emission perturbations (Figure S3). The BC-on-snow SRF exerted by the
European Winter perturbation is, at 17 mW m$^{-2}$ (Tg[C] yr$^{-1}$)$^{-1}$, 3.4 times stronger than for the
East Asian Winter perturbation and 2.4 times stronger than for the Global Winter
perturbation, also because of Europe's proximity to the Arctic. Jiao et al. (2014) assessed an
offline land surface model with BC deposition rates simulated by AeroCom models, including
OsloCTM2. They find that OsloCTM2 is among the models that overestimate BC-in-snow
amounts the most compared to measurements in the Arctic, suggesting a possible
overestimation of BC-on-snow SRF in this study.
BC aerosols are unusual among NTCFs because their strong absorption of shortwave
radiation is expected to trigger strong rapid adjustments (Koch and Del Genio, 2010), which
have been observed in marine stratocumulus regimes (Brioude et al., 2009; Wilcox, 2010).
Quantifying those rapid adjustments on a global scale is a challenge, because they are small
compared to internal variability in cloud fraction and top-of-atmosphere radiative fluxes.
Here, the quantification is done by prescribing control and perturbed distributions of BC
mass-mixing ratios simulated by OsloCTM2 into the Community Earth System Model





(CESM) version 1.0.4 (Neale et al., 2010). 30-year CESM simulations use fixed sea-surface
temperatures in order to suppress the long-term climate response and isolate the ERF. RFari
was quantified using multiple calls to the radiation scheme, following Ghan (2013). Because
aci are not included in the CAM4 atmospheric component of the CESM, the rapid
adjustments from the semi-direct effects of BC are calculated by subtracting its RFari from
total ERF. The reference CESM simulation uses BC concentrations taken directly from the
reference OsloCTM2 simulation. Then, to improve the signal-to-noise ratio between ERF and
unforced variability in perturbation simulations, the changes in BC are scaled before being
prescribed in CESM, following the equation:
$\qquad BC_{CESM} = (BC_{REF} - BC_{PERT}) * S + BC_{REF}$ $\qquad$ (Eq. 2)
where $BC_{CESM}$ are the distributions of BC concentrations prescribed into CESM, $BC_{REF}$ and
$BC_{PERT}$ are OsloCTM2's reference and perturbed distributions, respectively, and S is a scaling
factor. Table 6 gives the scaling factors imposed. The smaller the perturbation, the larger the
scaling factor required. Therefore, European perturbations are scaled by a factor 500 but
RotW perturbations only require a scaling factor of 30. The application of such large scaling
factors requires that rapid adjustments from the semi-direct effect scale linearly with the BC
perturbation imposed. This has been checked by imposing increasing scaling factors of 15, 50,
150, and 1500 to the East Asian Summer perturbation. Corresponding semi-direct SRFs are
$-44 \pm 121$, $-38 \pm 40$, $-38 \pm 12$, and $-35 \pm 1$ mW m$^{-2}$ (Tg[BC] yr$^{-1}$)$^{-1}$), indicating a
satisfactory level of linearity and supporting the application of large scaling factors.
Table 6 also gives the statistics of the resulting semi-direct SRFs taken over the 30 years of
simulation. With the exception of the East Asian Winter perturbation, semi-direct SRF is
negative, thus opposing the positive BC RFari. Semi-direct SRFs are weaker in Winter than in
Summer perturbations, as expected from a mechanism driven by absorption of shortwave
radiation. There are no strong regional variations in semi-direct SRFs. In spite of the large
scaling factors imposed, statistics are fragile and 90%-confidence intervals include 0 mW m$^{-2}$
for Winter perturbations. It is therefore important to keep in mind that the semi-direct
component of BC SRF is even more uncertain than the other components, and may not be
significantly different from zero.



## 3.3 Organic carbon aerosol

Figure 5 shows globally- and annually-averaged SRF for perturbations to primary organic carbon aerosol emissions in the four ECLIPSE models. OC SRF ranges from +1.2 mW m$^{-2}$ (Tg[C] yr$^{-1}$)$^{-1}$ for the East Asian Winter perturbation by ECHAM6 to -32.5 mW m$^{-2}$ (Tg[C] yr$^{-1}$)$^{-1}$ for the European Summer perturbation by NorESM1. ECHAM6 is again consistently associated with weak SRF because it only diagnoses ari. NorESM1 and OsloCTM2 disagree on the fraction of total RF contributed by ari. In OsloCTM2, ari dominates by contributing 77-83% of total SRF for European and East Asian perturbations and 63-67% for the Global perturbation. Winter perturbations are associated with slightly larger contributions of ari. In contrast, NorESM1 simulates a domination of aci, with ari only contributing 11-31% of total SRF for European and East Asian perturbations and 22-23% for the Global perturbation. Differences between seasons and regions are also more pronounced than in OsloCTM2. As for BC aerosols, changes in primary OC emissions are also associated with ozone and methane RF because of changes in the surface of aerosols available for heterogeneous reactions, but the resulting RFs are negligible and not reported in this study.

OC RF extends to larger areas (Figure S4) for models with longer lifetimes (Table 5), although the correlation between lifetime and SRF is not as clear as for SO$_2$ perturbation simulations. All models agree that OC SRF is stronger for Summer than Winter perturbations, which is expected for shortwave aerosol radiative effects. In addition, the four models suggest that European perturbations exert stronger SRFs than East Asian perturbations, which can be expected from a less polluted baseline in Europe.

OC aerosol surface concentrations are generally underestimated in Europe (Schulz et al., 2015) and at urban, remote, and marine sites worldwide (Tsigaridis et al., 2014). Those underestimations are attributed to underestimated sources in models, including primary emissions and secondary aerosol formation. Those underestimations may however not bias the SRF published in this study for the reasons discussed in Sect. 2.1.

Like sulphate aerosols, OC aerosols may exert rapid adjustments through changes in cloud microphysics. However, confidence in the ability of global models to represent those mechanisms with fidelity is low (Stevens and Feingold, 2009), so those adjustments are not quantified here.





## 3.4 Ammonia

Ammonia ($NH_3$) is, with nitrogen oxides, the precursor to ammonium nitrate aerosol. Ammonia perturbations have only been simulated by OsloCTM2 because it is the only participating model that represents the equilibrium between nitric acid, which is in the gas phase, and nitrate aerosols. Globally- and annually-averaged SRF for ammonia perturbations are shown in Figure 6. OsloCTM2 attributes 72 to 93% of SRF to ari, with larger fractions for Winter perturbations. In spite of nitrate aerosols having similar optical and cloud nucleus properties as sulphate aerosols, their simulated SRFs are about 10 times weaker at $-1$ mW m$^{-2}$ (Tg[$NH_3$] yr$^{-1}$)$^{-1}$. This weakness is due to two factors. First, formation of ammonium nitrate competes against that of ammonium sulphate, which is favoured by its better thermodynamic stability (Metzger *et al.*, 2002). The efficiency of nitrate precursor reductions therefore depends on regional sulphur dioxide levels. Second, nitrate aerosols are semi-volatile and dissociate back into the gas phase when temperatures increase. Nitrate aerosol formation is therefore hindered during daytime (Dall'Osto *et al.*, 2009), decreasing the ability of nitrate aerosols to interact with radiation. Sulphate aerosols have a more stable diurnal cycle, maximising their radiative forcing efficiency. Figure S5 shows distributions of nitrate RF simulated by OsloCTM2, which remains located near the perturbed regions, suggesting a short lifetime. That lifetime has however not been quantified in the model.

In AeroCom simulations of industrial-era RFari, 8 models with nitrate representations produced estimates of nitrate RF efficiency that range from 60 to 160% of the 8-model median of $-155$ W g[$NO_3$]$^{-1}$ (Myhre *et al.*, 2013b). Diversity in SRF will be at least as large, with diversities in lifetimes and aerosol-cloud interactions also contributing. Modelled nitrate lifetimes reported for present-day conditions in the literature indicate that this source of diversity is likely to be sizeable, with Bellouin *et al.* (2011) obtaining 3.1 days and Hauglustaine *et al.* (2014) having 4.6 days (50% longer). Diversity to aerosol-cloud interactions is most probably similar to the 10% reported for sulphate aerosols in Sect. 3.1. Overall, a conservative estimate of diversity is a factor of 2 each side of the OsloCTM2 estimate of nitrate SRF given here.





**3.5 Methane**
As discussed in Sect. 2, methane perturbations have been applied globally and annually,
instead of regionally and seasonally. This simplification is motivated by technical
considerations, because the long lifetime of methane would necessitate long model spin-ups.
However, this long lifetime also means that methane is relatively well mixed in the
atmosphere compared to shorter-lived species, thus the regional and seasonal nature of
perturbations are quickly lost, all perturbations converging into similar SRFs. This
convergence has been checked in NorESM1, where Summer and Winter perturbations have
been applied and found to yield methane SRFs that differ by only 7%.
The SRF exerted by methane itself is computed analytically on a global average in a four-
stage calculation.
- First, the methane feedback factor $f$ is derived from each model using Eq. 2 and 3 of
Stevenson *et al.* (2013), which require the knowledge of control and perturbed methane
burdens, and total methane lifetime $\tau_{tot}$. Following Stevenson *et al.* (2013), $\tau_{tot}$ accounts for
three methane sinks: destruction by OH, which is diagnosed in each model; losses to the
stratosphere, with a lifetime taken at 120 years; and losses to the soils, with a lifetime taken at
160 years. Those two lifetimes are also taken from Stevenson *et al.* (2013). ECLIPSE
feedback factors range from 1.28 to 1.46 (Table 7), in close agreement with the multi-model
mean derived by Holmes et al. (2013).
- In a second step, the equivalent methane emission perturbation $\Delta E$ is computed as
$$\Delta E = \Delta B / (f * \tau_{tot}) \qquad \text{(Eq. 3)}$$
where $\Delta B$ is the change in burden between the control and perturbed simulations.
- The third step computes methane RF in each model by inserting control and perturbed
methane volume mass-mixing ratios in the formula established by Myhre et al. (1998). The
mass-mixing ratio of nitrous oxide ($N_2O$), which is required to apply the formula, is taken at
325 ppb (WMO, 2014).
- Finally, methane SRF is computed as the RF divided by $\Delta E$, and increased by 15% to
represent the increase in stratospheric water vapour that follows methane oxidation (Myhre et
al., 2007b).





Methane burdens, lifetimes, and all the global averages involved in computing the methane
contribution to total methane SRF in the three ECLIPSE models, are given in Table 7. It is
important to note that the diversity in modelled methane SRF is not due to uncertainties in the
radiative properties of the molecule, but rather due to the diversity in simulating present-day
burdens, which affect the baseline of a non-linear RF.
In addition to the SRF exerted by methane itself, components due to perturbations to aerosols
and ozone precursors contribute to total methane SRF (Figure 7). Aerosol and ozone RFs are
derived using the methods described in Sect. 2.
The aerosol component of methane SRF arises from the increase in OH that follows the
decrease in $CH_4$ concentrations. This increase promotes $SO_2$ oxidation into sulphate aerosols,
contributing a negative RF that, once divided by the negative methane emission change,
translates into a positive contribution to total SRF. That contribution is very diverse among
models, varying from a weakly negative contribution in OsloCTM2 to a strongly positive
contribution in HadGEM3. The OsloCTM2 value is from a simplified calculation, using
distributions of radiative forcing efficiencies instead of the full radiative transfer calculations
used by the other models and for the other perturbations. This simplified calculation only
represents ari, and the OsloCTM2 aerosol contribution would likely be positive, like the other
models, if aci were included. But three other aspects of the models contribute to the diversity
in estimates of aerosol contributions to methane SRF. The first aspect is the size of the
relative increase in global OH burden that follows the decrease in methane concentrations
imposed in ECLIPSE. NorESM1 and OsloCTM2 obtain similar global averages, at +4.5%
and +4.6%, respectively, but HadGEM3 simulates a larger sensitivity of OH, at +7%.
Secondly, other limitations restrict the aerosol response in some models, but not others. For
example, NorESM simulates aerosol SRFs of differing signs (Figure S6), which indicate
different responses of local chemistry, possibly mediated by changes in oxidation pathways
by $O_3$ and $H_2O_2$. In HadGEM3 however, aerosol SRF is uniformly positive across the globe
(Figure S6), indicating that once OH is increased, no further limitation restricts the size of the
aerosol response. The realism of those responses are difficult to confirm from observations, as
evidence for changes in the oxidising capacity of the atmosphere are lacking. Thirdly, the
inclusion of nitrate aerosols in OsloCTM2 counteracts the sulphate aerosol response, because





increases in ammonium sulphate aerosol formation are detrimental to ammonium nitrate
aerosol formation.
In stark contrast to the diversity seen in the aerosol component of total methane SRF, all three
models simulate ozone contributions to methane SRF close to one third of the SRF of
methane itself. This chemical feedback is therefore in good agreement among models, and is
proportional to the size of the methane perturbation. Figure S7 shows that the models also
agree well on the geographical distribution of the ozone SRF, with a maximum at the tropical
boundaries.
**3.6  Nitrogen oxides**
Figure 8 shows globally- and annually-averaged SRF for nitrogen oxide perturbations in the
three ECLIPSE models with tropospheric ozone chemistry schemes. Total $NO_X$ SRF varies
from $-0.16$ mW m$^{-2}$ (Tg[$NO_2$] yr$^{-1}$)$^{-1}$ for the East Asian Summer perturbation with
NorESM1 to $-1.97$ mW m$^{-2}$ (Tg[$NO_2$] yr$^{-1}$)$^{-1}$ for the Global Winter perturbation with
OsloCTM2. SRF components are region- and season-dependent, but the dependence of net
SRF is less pronounced because the short-lived ozone and aerosol contributions compensate
each other. Total $NO_X$ SRF is negative in all models for all regions and seasons, but results
from the addition of negative contributions by methane and primary-mode ozone SRF and
positive contributions by short-lived changes in ozone. Models disagree on the sign of the
aerosol contribution. Quantitatively, the three models are in good agreement on the SRF
exerted directly by ozone, both on a global average (Figure 8) and on patterns (Figure S9) but
disagree on the contributions of methane/primary-mode ozone and aerosols.
The methane and primary-mode ozone SRF are calculated as global averages only, by
multiplying the change in methane burden due to its reaction with OH by a radiative forcing
efficiency (RFE). Methane RFE is taken at 0.363 mWm$^{-2}$ ppbv$^{-1}$ (Table 8.A.1 of Myhre et al.,
2013a). Primary-mode ozone RFE is computed as the ratio of ozone RF to total methane
burden change in the methane perturbation simulations (see Sect. 3.5). That RFE is more
easily expressed as a fraction of the methane RFE, with good agreement among ECLIPSE
models: 0.396 for HadGEM3, 0.385 for NorESM1, and 0.395 for OsloCTM2. Figure 8 shows
that the resulting methane and primary-mode ozone SRF are in good agreement between



OsloCTM2 and NorESM1 for NOX perturbations, but HadGEM3 simulates a weaker SRF.
This is consistent with results from the methane perturbation (Table 7).
Models strongly disagree on the aerosol contribution to $NO_X$ SRF. That contribution is
generally negative, but NorESM1 also simulates positive contributions, especially in Winter
perturbations. In an absolute sense, the strongest contributions are simulated by OsloCTM2
for Europe and East Asian Summer perturbations, and by HadGEM3 for East Asian Winter
and Global perturbations. Figure S8 shows that those disagreements stem from differences in
regional responses. Both HadGEM3 and NorESM1 show positive aerosol RFs centred on the
regions being perturbed, caused by a decrease in sulphate aerosol formation through OH
oxidation because OH levels are decreased. The $SO_2$ not oxidised and not deposited is
transported downwind of the perturbed region, where it promotes sulphate aerosol formation
in the absence of oxidant limitation: in those regions, both models simulate negative aerosol
RFs. The balance between regions of positive and negative aerosol RF varies depending on
the model, the perturbed region, and the season. In contrast, OsloCTM2 does not simulate this
dipole of responses: its aerosol contribution is negative almost everywhere on the globe. The
representation of nitrate aerosols explains that difference of behaviour compared to the other
models. Nitrate exerts between 50 and 95% of RFari to $NO_X$ perturbations in OsloCTM2,
with largest contributions in Northern Hemisphere winter months, adding a negative RF in,
and downwind of, the perturbed regions. This brings the total aerosol SRF firmly into
negative values.
**3.7   Volatile organic compounds**
Figure 9 shows globally- and annually-averaged SRF for volatile organic compound
perturbations in the three ECLIPSE models with tropospheric ozone chemistry schemes. Total
VOC SRF varies from –0.53 mW m$^{-2}$ (Tg[C] yr$^{-1}$)$^{-1}$ for the Europe Winter perturbation with
NorESM1 to +2.07 mW m$^{-2}$ (Tg[C] yr$^{-1}$)$^{-1}$ for the same perturbation with HadGEM3. So
models disagree on the sign of total VOC SRF, although it is generally positive. Qualitatively,
all models agree that ozone, methane, and primary-mode ozone contribute positive SRFs. So
the qualitative disagreement stems from the sign and magnitude of the aerosol contribution.
Methane and primary-mode ozone SRFs are computed as described in Sect. 3.6.



VOC chemistry is particularly complex and diverse. Decreasing VOC leads to a decrease in
their oxidation products, CO and $O_3$, therefore increasing OH and decreasing $CH_4$
concentrations (Lin et al., 1988). Different VOCs have different photochemical $O_3$ creation
potentials (Derwent et al., 2001). The three ECLIPSE models include a different number of
VOC species. The model with the largest number of VOC species is OsloCTM2, with 40
species: 28 in the tropospheric chemistry scheme and 12 in the secondary organic aerosol
scheme. Its broader range of VOC lifetimes and ozone production potentials means that it
simulates the strongest ozone SRF. HadGEM3 is at the other end of the range of species
considered and simulates the weakest ozone SRF. The models agree better in terms of
simulated patterns of ozone RF, which is mostly located in the latitude band of the perturbed
region (Figure S11).
Models disagree in the sign and strength of the aerosol contribution to VOC SRF. HadGEM3
simulated positive SRFs to all perturbations, except for East Asian Winter. Compared to the
ozone precursor perturbations discussed previously, which impact sulphate and nitrate
formation, VOC perturbations introduce a new way of perturbing aerosols, via secondary
organic aerosol formation. The strength of this link varies strongly between models because
of the heterogeneity in the number and type of VOC represented. Although HadGEM3 agrees
with NorESM1 and OsloCTM2 that aerosol RF is negative above the perturbed regions
(Figure S10), those negative RFs are weak and therefore easily compensated on a global
average by noisy positive contributions in regions where the aerosol internal mixture has been
perturbed (e.g. north-western Russia, Indonesia, South America). Observational constraints on
such internal mixture perturbations are lacking, so it is not currently possible to assess the
realism of HadGEM3's response. Aerosol SRF in OsloCTM2 is weaker than that in
NorESM1. This is due to the representation of nitrate aerosols, which counteract part of the
RF exerted by changes in sulphate aerosols, but also to a weaker RFaci contribution.

### 3.8   Carbon monoxide

Figure 10 shows globally- and annually-averaged SRF for carbon monoxide perturbations in
the three ECLIPSE models with tropospheric ozone chemistry schemes. Total CO SRF varies
from +0.18 mW $m^{-2}$ (Tg[CO] $yr^{-1}$)$^{-1}$ for the Europe Summer perturbation with NorESM1 to





+0.26 mW m$^{-2}$ (Tg[CO] yr$^{-1}$)$^{-1}$ for the East Asia Summer perturbation with HadGEM3. There
are no strong differences in SRF between the different source regions. Seasonally however,
models suggest the increased methane contribution in Winter perturbations make those more
efficient at exerting an SRF than Summer perturbations. Models agree that East Asian
perturbations exert slightly stronger SRFs than European perturbations because of a stronger
SRF by ozone. East Asian ozone exerts a stronger RF per unit ozone burden because of higher
NO$_X$ background in that region but also because it is closer to the Equator, where more
sunlight leads to a more active photochemistry (Berntsen et al. 2006). Qualitatively, all
models agree that ozone, methane, and primary-mode ozone contribute positive SRFs.
Aerosol RF is also positive in HadGEM3 and NorESM1, but is weakly negative in
OsloCTM2. Methane and primary-mode ozone SRFs are computed as described in Sect. 3.6.
The contributions by short-lived perturbations to ozone are in reasonable agreement among
the models, both from a global average (Figure 10) and pattern (Figure S13) point of view.
All models also agree that the methane SRF contribution is larger than that of short-lived
ozone changes. This is in contrast to the results for VOC (Sect. 3.7), where the ozone
contribution dominates, and stems from the fact that CO has a weaker ozone production
potential caused by slower reaction rates (e.g. Bowman (1995)).
Both NorESM1 and OsloCTM2 simulate relatively weak contributions of aerosols to CO
SRF. The contribution simulated by OsloCTM2 is negative because the positive RFs exerted
by sulphate and secondary organic aerosols are more than compensated by a negative RF by
nitrate aerosols. HadGEM3 simulates a relatively strong response of aerosols to CO
perturbations (Figure S12), but that is because biomass-burning emissions were also perturbed
in this model. NorESM1 and OsloCTM2 only perturbed fossil-fuel combustion emissions,
and the results suggest that links between CO and aerosols are stronger for biomass-burning
sources. However, other sources of diversity, including the representation of atmospheric
chemistry, could also explain the differences in behaviour between HadGEM3 and the other
participating models.



## 3.9   Shipping sector
Figure 11 shows globally- and annually-averaged SRF for all species ($SO_2$, BC, OC, $CH_4$,
$NO_X$, VOC, and CO) emitted by the shipping sector. ECHAM6 lacks a tropospheric ozone
chemistry scheme, and therefore only simulates the aerosol contribution, and diagnoses RFari
only. OsloCTM2 is the only model that includes BC-on-snow RF and that quantifies BC
semi-direct RF (see methods in Sect. 3.2). Total shipping SRF varies from –6.7 mWm$^{-2}$ (Tg
yr$^{-1}$)$^{-1}$ for the Summer perturbation by NorESM1 to –1.47 mWm$^{-2}$ (Tg yr$^{-1}$)$^{-1}$ for the Winter
perturbation by ECHAM6. Models agree qualitatively that ozone contributes a positive SRF.
Aerosols provide a negative SRF. Methane and primary-mode ozone also contribute a
negative SRF, mainly driven by emissions of $NO_X$. The qualitative agreement for those
components is also good. Methane and primary-mode ozone SRFs are computed as described
in Sect. 3.6.
The SRF contributed by short-lived changes in ozone are in good agreement among models,
both in terms of global averages (Figure 11) and geographical patterns (Figure S16), with
maxima in the Tropics. ECLIPSE models may overestimate that contribution, however,
because they do not represent ship plumes but instead assume instantaneous dilution of
emissions over their grid boxes. This assumption is known to lead to an overestimate of ozone
production by $NO_X$ (Paoli et al., 2011). BC-on-snow (see also Figure S15) and BC semi-direct
SRF, which are quantified from OsloCTM2 simulations as described in Sect. 3.2, are weak.
Methane SRF is a large contribution to shipping SRF because ships emit in pristine
environments, where ozone precursor emissions have a relatively larger impact than in
polluted regions.
Models agree that aerosols dominate shipping SRF, but disagree on the strength of that
contribution. The causes of the disagreement are a convolution of the disagreements listed
above for individual species: among those, different lifetimes, different strengths of RFaci,
and different treatment of the aerosol mixing state. Geographical patterns are similar among
models and reflect main shipping routes (Figure S14). NorESM1 shows a region of positive
aerosol RF in the Arctic, caused by the long-range transport of its long-lived BC. That model
overestimates BC Arctic concentrations in the Summer, however (Sect. 3.2), so the positive
RF region may not be realistic.



## 4 Matrix of specific radiative forcing

Section 3 described the results of the perturbation simulations for each NTCF and each model. This section describes how those results can be summarised into a best estimate and range, which are more useful for most applications. However, all global numbers are given in Supplementary Materials, to allow users to make other choices in deriving a best estimate or range.

For each regional and seasonal perturbation by an NTCF, best estimates of SRF are provided for each RF mechanism: aerosols (sum of RFari and RFaci), BC deposition on snow, BC rapid adjustments to semi-direct effects, short-lived changes in tropospheric ozone concentrations, methane, and primary-mode ozone. The best estimate on net SRF is the sum of the best estimates of all RF mechanisms that are relevant to the NTCF considered. Inter-model diversity is represented by an interval ranging from the weaker SRF, obtained by adding the weaker estimates of all RF mechanisms, to the stronger SRF, obtained by adding the stronger estimates of all RF mechanisms. Best estimates of RF of BC deposition on snow and BC rapid adjustments from semi-direct effects are available from only one model, so are also taken to represent high and low estimates. It is however important to note that the statistics on BC adjustments from semi-direct effects are not robust and that it may in fact not be significantly different from zero for the Winter perturbations, as discussed in Sect. 3.2.

It can be argued that models that fail to provide realistic simulations of key aspects of NTCF distributions and RF mechanisms should be discarded. For example, Shindell et al. (2013) screen the 10 models that participated in ACCMIP for their ability to reproduce observed total AOD and its recent trend, leading to a reduction in inter-model diversity. Such a screening is not applied here because models do not exhibit uniform skill at reproducing aerosol or ozone distributions: a model that could be considered best in one region often shows poorer skill in another.

Nevertheless, decisions are required here on the inclusion of models that do not diagnose RFaci, or simulate long BC lifetimes, or lack nitrate aerosols, or simulate complex aerosol-chemistry responses. The decisions are:

- For RFaci, ECHAM6 is not included in best estimates of aerosol SRF because it does not diagnose aci, which according to the other models is an important, and indeed often dominant,



contribution of total aerosol RF. It is possible that RFaci is in fact compensated by rapid
adjustments in cloud liquid water path (e.g. Christensen and Stevens, 2011), meaning that
ECLIPSE models overestimate the strength of aerosol SRF. However, there is currently no
evidence that such compensation happens on a global scale.
- For BC lifetimes, a possible decision would be to discount models with BC lifetimes longer
than about 4 days, which is the lifetime obtained by constraining BC mass concentration
profiles with aircraft observations (Wang et al., 2014b; Hodnebrog et al., 2014). That decision
would give more weight to the aerosol SRF simulated by ECHAM6 and HadGEM3.
However, comparisons to surface observations in the Arctic suggest that ECHAM6 and
HadGEM3 underestimate BC concentrations in that region (Sect. 3.2), perhaps because
aerosols do not stay long enough in the atmosphere to be transported to the Arctic in those
two models. Reconciling mixed conclusions from different indirect observational constraints
on lifetime is therefore warranted. In the meantime, no model is discounted in this study when
producing the best ECLIPSE model estimate and range of BC SRF. Still, the tendency of
models to put BC too high in the atmosphere needs to be kept in mind, as it leads to an
overestimated SRF.
- For nitrate, the descriptions of results for the $SO_2$ (Sect. 3.1) and ozone precursor
(Sect. 3.6—3.8) perturbations note the importance of co-variations in nitrate aerosols. Those
are only represented in OsloCTM2 but are crucial in that model in determining the strength,
and on occasions even the sign, of aerosol SRF. For that reason, it is decided here to add the
nitrate SRF simulated by OsloCTM2 to the aerosol SRF of the other models. This solution is
crude, as it is known that model diversity in simulating nitrate distributions is large (Myhre et
al., 2013b) and a correlation between sulphate and nitrate RF can be expected from their links
through ammonium. But in the absence of a solid understanding of those correlations, the
solution adopted here has the merit of simplicity and prevents misleading overcorrections.
- For aerosol-chemistry interactions, HadGEM3 simulates complex responses of aerosols to
ozone precursor perturbations. This is particularly true of VOC perturbations (Sect. 3.7),
where HadGEM3 simulates a positive SRF when NorESM1 and OsloCTM2 agree on a
negative contribution. At this stage, the realism of HadGEM3's response cannot be confirmed
by observations, but nor can it be challenged. It is decided for that reason to include



HadGEM3 in the best estimate and range of VOC, with the caveat that the model behaviour is
peculiar.
Figure 12 shows the resulting best SRF estimate for all perturbations. Best estimates for each
mechanism are shown in colour. Best estimates for the net SRF are shown as black bars, with
the range from weaker to stronger estimates represented as whiskers. The range for $NH_3$
perturbations, which have been quantified from one model only, is assumed to be a factor 2
(Sect. 3.4). Model diversity ranges are often sizeable, but rarely include zero, indicating that
models generally agree on the sign of the SRF of a given NTCF. $SO_2$, OC, $NH_3$, $NO_X$, and
shipping sector perturbations exert negative SRF. BC aerosols, methane and CO exert positive
SRF. The sign of the SRF exerted by VOC perturbations is less robust. Its best estimate is
positive, but models cannot agree on the sign and the diversity range is large. The sign of
VOC SRF depends on the strength and sign of aerosol responses, including secondary organic
aerosols.
Quantitatively, best estimates of BC SRF are the strongest of all NTCFs, even after
accounting for rapid adjustments from semi-direct effects. Aerosol SRFs are generally
stronger than ozone precursor SRFs, with the exception of $NH_3$ perturbations, which exert
weak SRF because of competition with ammonium sulphate aerosol formation and because
the diurnal cycle of nitrate aerosol formation is unfavourable to ari (Sect. 3.4). $NO_X$ exerts the
strongest SRF of all ozone precursor perturbations, although VOC perturbations are
potentially as strong. Shipping SRF is strong because of strong contributions by aerosols and
methane.
The best estimates of this study are included in Table 1 for convenient comparison to previous
studies. This study suggests a revision towards stronger SRFs for $SO_2$ and OC perturbations
because of the inclusion of RFaci. In contrast, this study's BC SRF is not very different than
that derived by studies that consider ari only, because the inclusion of aci, deposition on
snow, and rapid adjustments from semi-direct effects contributes only a weakly positive, and
even at times negative, SRF. The BC SRF in this study is however several times weaker than
that proposed by Bond et al. (2013), primarily because modelled ari is not scaled up to correct
for perceived underestimations in absorbing aerosol optical depth. Such corrections have been
challenged by Wang et al. (2014a) and Samset et al. (2014), so caution is justified. For
methane and ozone precursor perturbation, the study agrees well with previous efforts in



estimating the methane contribution. The SRF exerted by short-lived perturbations to ozone
concentrations is generally revised upward. This study quantifies aerosol responses to ozone
precursor perturbations for the first time across multiple models. Aerosols contribute a
negative SRF to NOx perturbations, positive to CO perturbations. The contribution of
aerosols to VOC SRF is negative in two models, but positive in another. Finally, the study
also singles out the shipping sector, finding that its SRF is negative and mostly contributed by
aerosols and methane.

## 9    4.1   Seasonality

For all perturbations, SRF best estimates are given for emission reductions applied in two
periods, May-Oct and Nov-Apr, which are labelled in Figure 12 Summer and Winter,
respectively, because emission perturbations are predominantly located in the Northern
Hemisphere. The seasonality of methane perturbations was not considered because the time of
emission becomes quickly irrelevant compared to the long residence time of methane in the
atmosphere. NorESM1, which implemented Summer and Winter methane perturbations,
confirms that seasonality of methane emissions has a small impact, making a 7% difference in
methane SRF.
For aerosol primary and precursor perturbations, which are largely located in the Northern
Hemisphere, Summer emission reductions exert strong SRFs because the RF mechanisms act
mostly on shortwave radiation. For RFari, anthropogenic aerosols are predominantly located
in the accumulation mode, at sizes which interact most efficiently with shortwave radiation.
For RFaci, changes to cloud albedo operate in the shortwave spectrum only, although BC
semi-direct SRF has a longwave component. In addition to RF mechanisms, chemical
production and sinks (mainly from precipitation) also influence seasonality. $SO_2$ photolysis is
an example of a reaction favoured by higher, summertime, shortwave radiative fluxes.
Temperature is also a factor, especially in nitrate aerosol formation, which is favoured by
colder temperatures. This dependence explains the unusual seasonality of $NH_3$ perturbations,
which exert stronger SRFs in Winter perturbations for East Asian and on a global average.
The fact that European perturbations behave differently is linked to the lower sulphate aerosol



levels in Europe, reducing their ability to limit nitrate formation in both summer and winter
months.
The SRF of ozone precursor perturbations is exerted across both the shortwave and longwave
spectra, so its seasonality is not as strong as for aerosol perturbations and the details of ozone
formation pathways are important. Figure 12 shows that Winter $NO_X$ perturbations exert
stronger SRFs, except for European perturbations. The seasonality of $NO_X$ RF depends on the
level of cancellation between the positive ozone contribution and the negative methane
contribution. Derwent et al. (2008) found using a CTM that there are no simple relationships
that explain that competition, which also varies regionally. Our results replicate that
complexity. CO Winter perturbations are consistently stronger than Summer perturbations,
but differences are generally small. Finally, VOC perturbations may have a seasonality where
Summer perturbations are stronger than Winter perturbations, but model diversity is large so
the seasonality is uncertain.

## 4.2 Latitudinal variations

Figures 13a and 13b show best estimates and ranges of SRF for aerosols and ozone
precursors, respectively, across four latitude bands: 90N—60N, 60N—28N, 28N—28S, and
28S—90S. Those bands have been chosen to represent the Arctic, mid-latitudes, Tropics, and
Southern Hemisphere extratropical latitudes, respectively. The Southern Hemisphere is less
resolved than the Northern Hemisphere because anthropogenic emissions are predominantly
located in the latter. European emission perturbations are entirely located in the second band
(60N—28N). East Asian emission perturbations also include the northern portion of the third
band (28N—28S). RotW and shipping perturbations are located across all four bands, but
again with Northern Hemisphere emissions having more weight.
Latitudinal averaging of RF is done on the annual distributions shown as Supplementary
Figures. SRF is then computed by normalising by the globally-averaged emission change: so
for a given perturbation, both global and latitudinal SRFs share the same normalisation
factors. Annual distributions are however not available for methane RF and BC rapid
adjustments to semi-direct effects. Methane RF has been computed as a global average only
(see Sect. 3.5). It is assumed here to be uniformly distributed across globe, which is justified



on an annual basis by the well-mixed nature of methane. BC rapid adjustments are associated
with noisy distributions (see Sect. 3.2), so there is low confidence in the significance of
regional patterns. They are assumed here to follow the same latitudinal distribution of BC
RFari, which is justified by the close physical links between the two RF processes.
Figures 13 show that although SRF is typically stronger in the latitude band where the
emission perturbation is applied, it is not confined to that latitude band. This behaviour is
expected from atmospheric transport, and has been found previously in other modelling
studies (e.g. Shindell and Faluvegi, 2009). European aerosol and precursor perturbations
affect the Arctic in a sizeable way. The BC European and Global Winter perturbations may
even exert a stronger positive SRF in the Arctic than in mid-latitudes where the perturbations
are located, because of the added positive contribution of BC-on-snow RF. The SRF exerted
by East Asian perturbations is more confined to mid-latitudes, because atmospheric transport
preferentially advects the perturbations towards the Pacific Ocean rather than the Arctic,
especially in Winter perturbations (Figure S2).
Ozone precursor perturbations (Figure 13b) tend to be more diffuse than their aerosol
counterparts, in part because of the longer lifetime of ozone in ECLIPSE models (Table 5) but
also because perturbations to OH lifetime are more efficient in the Tropics (Berntsen et al.,
2006). SRF of ozone precursor perturbations are therefore strong in both Northern
Hemisphere mid-latitudes, where the perturbations are located, and the Tropics. For European
and East Asian perturbations, the Arctic is generally associated with weaker SRFs, except for
CO, which is associated with more spatially uniform SRFs because methane RF is the main
contributor. The SRF of shipping sector perturbations peaks in the Northern Hemisphere,
where most shipping lanes are located.

## 5   Conclusion

This study provides NTCF SRFs by using ECLIPSE model simulations by four general
circulation and chemistry-transport models: ECHAM6, HadGEM3, NorESM1, and
OsloCTM2. SRFs are given for eight NTCF, four regions or sectors, and six RF mechanisms.
The four regions are Europe, East Asia, global average, and the shipping sector. The eight
NTCFs or NTCF precursors are $SO_2$, BC, OC, $NH_3$, methane, $NO_X$, CO, and VOC. $NH_3$
perturbations were applied in OsloCTM2 only, which includes a representation of nitrate



aerosols. The six RF mechanisms are aerosols (both ari and aci), BC deposition on snow, BC
rapid adjustments from semi-direct effects, short-lived ozone changes, methane, and primary-
mode ozone. OsloCTM2 is the only model used to estimate BC deposition on snow and BC
rapid adjustments from semi-direct effects. ECHAM6 does not simulate ozone chemistry, so
does not provide SRFs for the last three RF mechanisms on the list.
Models generally agree on the sign of the total SRF of a given NTCF. The SRF exerted by
$SO_2$, OC, $NH_3$, $NO_X$, and shipping sector perturbations is negative. The SRF exerted by
methane and CO is positive. Models also agree that BC SRF is positive, but is weakened by
rapid adjustments from semi-direct effects. Models do not agree on the sign of VOC SRF,
although its best estimate is positive, because of diversity in the aerosol responses. Model
diversity has multiple and complex roots, but four important aspects stand out.
- Diversity in modelled NTCF lifetimes is large, with longest lifetimes being 1.5 to 2.5
times longer than the shortest lifetimes depending on NTCF. Differences in lifetime
affect both the reach of long-range transport and the reference baseline.
- The unperturbed baseline causes diversity for non-linear RF mechanisms, such as
RFaci and methane RF. It is also a common cause for regional differences in SRF.
- The number of species represented varies among models. Nitrate and secondary
aerosols modulate the strength of the SRF exerted by $SO_2$, $NO_X$, VOC, and CO
perturbations, but are not included in all models, causing potentially misleading results
in models where those aerosol species are absent. Models that include VOC emissions
also account for a different number and type of VOC species.
- Interactions between aerosols and chemistry, and particularly aerosol responses to
changes in the oxidising capacity of the atmosphere and secondary organic aerosol
formation, affect the strength, possibly even the sign, and the seasonality of SRF. The
strength of those interactions differs among models.
Harmonising modelling capabilities, and deriving observational constraints on modelled
lifetimes (e.g. Kristiansen et al., 2016) and responses of OH concentrations to chemistry
perturbations will be useful in reducing model diversity while also quantifying model skill at
simulating atmospheric composition with fidelity.  Other causes of diversity include different
aerosol optical properties, including BC absorbing properties (e.g. Myhre et al., 2013b);
different vertical profiles (e.g. Samset et al., 2013); different cloud processes, which affect the



strength of RFaci (e.g. Quaas et al., 2009); and host model considerations, such as the use of
different radiative transfer schemes (Stier et al., 2013) and different simulations of horizontal
and vertical cloud distributions.
Models agree well on the ranking of the SRF of the NTCF considered in this study, with SRF
exerted by aerosol perturbations being up to an order of magnitude stronger than methane and
ozone precursor perturbations. An exception to that observation is the SRF exerted by $NH_3$
perturbations, which is weakened by the diurnal cycle of nitrate aerosol formation and
competition from sulphate aerosols. In terms of best estimates, BC exerts the strongest SRF of
all aerosol and precursor perturbations, while $NO_X$ exerts the strongest SRFs of ozone
precursor perturbations, although VOC SRF may be as strong. Shipping sector SRF, which
combines strong contributions from aerosols and methane, is relatively strongly negative.
However, in terms of climate mitigation, this ranking of NTCF SRF should be understood in
the context of their individual emission rates. For example, although the BC SRF is about 20
times stronger on a global average than that of methane, the anthropogenic emission rate of
methane is 200 times larger. The negative RF obtained by the same relative change in
emission rates would therefore be stronger for methane than for BC.
Regionally, European aerosol perturbations exert stronger SRF than East Asian perturbations,
because East Asia has a more polluted baseline which saturates RFaci and dampens the
impact of emission reductions. The regional dependence of ozone precursor SRF is more
complex, and no systematic rule is found, in common to previous studies (Derwent et al.,
2008). The strong aerosol and methane RF exerted by shipping perturbations are likely due to
emitting in pristine environments. SRF has an unequal latitudinal distribution, generally
peaking in the latitude band where the perturbation is applied, although other regions, notably
the Arctic, are affected through changes in transported NTCFs. In that respect, BC Winter
perturbations are notable for exerting stronger positive SRF in the Arctic than in Europe
because of the added contribution of BC-on-snow RF.
Seasonally, RF mechanisms that act primarily on shortwave radiation, such as aerosol ari and
aci, make the SRF of Summer perturbations stronger than that of Winter perturbations.
However, that seasonality may be inverted for species, such as nitrate aerosols, for which
chemical production has itself a strong seasonal dependence. Seasonality of ozone precursor
perturbations is more complex and region-dependent, but also less pronounced than for



aerosols. NorESM1 simulations found that the seasonality of methane SRF is weak, as
expected from a relatively long-lived NTCF.
Ideally, the SRF matrix presented here should include rapid adjustments to all RF
mechanisms and be a matrix of specific ERF. But quantifying ERF is more challenging than
the already challenging task of quantifying RF, especially for the small regional and seasonal
perturbations considered here. The challenge is to distinguish, in a statistically robust way,
rapid adjustments from internal variability. The only rapid adjustment considered in this study
is from the semi-direct effect of BC aerosols, and the statistics are fragile. Nudging of
temperature and wind speeds have shown promise in decreasing the size of internal variability
(Kooperman et al., 2012), but whether that method also suppresses rapid adjustments is
unknown. One possible variation of that method is to allow temperature to adjust freely to
semi-direct effects, while wind speeds remain nudged to decrease internal variability between
perturbed and unperturbed simulations. Implemented in HadGEM3, that method successfully
reproduces the globally-averaged seasonality of ERF and subsequent precipitation changes
simulated by free-running simulations (Figures S17 and S18). The simulations required to do
so are 6 times shorter and have better statistics. This encouraging result holds for a variety of
RF mechanisms, including a doubling of carbon dioxide concentrations and RF ari and aci.
That method assumes that thermodynamical and dynamical responses are separated, at least
over rapid adjustment timescales. Whether that assumption is true and allows ERF to be
quantified confidently remains to be demonstrated.

**Data availability**

Supplementary Materials include spreadsheets giving all globally-averaged numbers for all
perturbation simulations and all radiative forcing mechanisms, by all models.

**Acknowledgements**

The research and simulations described in this study were funded by the European Union
Seventh Framework Programme (FP7/2007-2013) under grant agreement no 282688 –
ECLIPSE. The research team based at the University of Reading acknowledges use of the
MONSooN supercomputing system, a collaborative facility supplied under the Joint Weather



and Climate Research Programme, which is a strategic partnership between the UK Met
Office and the UK Natural Environment Research Council. The University of Leipzig team
acknowledges additional funding by the European Research Council (QUAERERE,
GA 306284) and computing time from the German Climate Computing Centre (DKRZ). The
CICERO team acknowledges additional funding from the Research Council of Norway
through the NetBC (no. 244141) and SLAC (no. 208277) projects.

**Author contribution**

N. Bellouin, G. Myhre, and J. Quaas designed the experiments as part of the ECLIPSE
project. N. Bellouin, L. Baker, Ø. Hodnebrog, D. Olivié, R. Cherian, C. Macintosh, B.
Samset, and A. Esteve ran the experiments or radiative transfer calculations, and analysed the
data sets. B. Aamaas provided additional data analysis in the perspective of climate metrics
users. N. Bellouin prepared the manuscript with contributions from all co-authors.

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




**6 Tables**
**Table 1.** *Specific radiative forcing (SRF), in mWm$^{-2}$ (Tg yr$^{-1}$)$^{-1}$, of near-term climate forcers,*
*as estimated by scientific assessments and multi-model inter-comparisons. Numbers shown*
*are median and full range for all studies, except for: - Bond et al. (2013), where best estimate*
*and 90% confidence range are given; - Yu et al. (2013), where mean and standard deviation*
*are given; - this study, where average and full range are given. Black Carbon (BC) and*
*Organic Carbon (OC) aerosols are for fossil- and bio-fuel sources only, except for Bond et*
*al. (2013) which also includes biomass-burning sources. For aerosols, the radiative forcing is*
*for aerosol-radiation interactions (ari) only, except for the estimate by Bond et al. (2013)*
*denoted "All", which also includes aerosol-cloud and aerosol-surface interactions, and for*
*estimates by this study, which also include aerosol-cloud interactions (aci).*

| Emitted compound | Climate Forcer | Reference | Method | SRF (mWm$^{-2}$ (Tg yr$^{-1}$)$^{-1}$) |
|---|---|---|---|---|
| SO$_2$ | SO$_4$ | Myhre et al. (2013b) | AeroCom, 15 models, ari only | −3.5 (−5.5 to −1.5) |
| | | Shindell et al. (2013) | ACCMIP, 9 models, ari only | −4.3 (−6.4 to −2.0) |
| | | Yu et al. (2013) | HTAP, 8 models, 4 source regions, ari only | −2.9 ± 0.8 to −3.9 ± 0.8 depending on region |
| | | *This study* | ECLIPSE, 3 models, 3 source regions, 2 seasons, ari +aci | −3.1 to −10.7 (−1.9 to −17.7) depending on region |
| OC | OC | Myhre et al. (2013b) | AeroCom, 15 models, ari only | −3.8 (−7.6 to −1.3) |
| | | Shindell et al. (2013) | ACCMIP, 4 models, ari only | −3.8 (−10.1 to −1.3) |
| | | Yu et al. (2013) | HTAP, 8 models, 4 source regions, ari only | −3.7 ± 1.8 to −4.4 ± 1.7 depending on region |
| | | *This study* | ECLIPSE, 3 models, 3 source regions, 2 seasons, ari +aci | −4.4 to −22.5 (+1.2 to −32.5) depending on region |
| BC | BC | Bond et al. (2013) | Assessment of models with observational constraints, ari only | +114.8 (+13.1 to +208.2) |



| | | | | |
|---|---|---|---|---|
| | | Bond et al. (2013) | Assessment of models with observational constraints, all RF mechanisms | +180.3 (+27.9 to +344.3) |
| | | Myhre et al. (2013b) | AeroCom, 15 models, ari only | +45.3 (+15.1 to +75.6) |
| | | Shindell et al. (2013) | ACCMIP, 5 models, ari only | +50.4 (+35.3 to +95.7) |
| | | Yu et al. (2013) | HTAP, 8 models, 4 source regions, ari only | +25.3±14.6 to +37.4±19.3 depending on region |
| | | *This study* | ECLIPSE, 4 models, 3 source regions, 2 seasons, ari+aci+deposition on snow and rapid adjustments from the semi-direct effect | +28.7 to +69.7 (+9.8 to +101.1) depending on region |
| $NH_3$ | $NO_3$ | Myhre et al. (2013b) | AeroCom, 5 models, ari only | –3.9 (–13.3 to –1.0) |
| | | *This study* | ECLIPSE, 1 model, 3 source regions, 2 seasons, ari+aci | –0.5 to –1.4 depending on region |
| $CH_4$ | $CH_4$ | Stevenson et al. (2013) | ACCMIP, 6 models | +2.2 (+1.8 to +3.0) |
| | | *This study* | ECLIPSE, 3 models | +1.5 (+1.2 to +2.0) |
| | $O_3$ | Stevenson et al. (2013) | ACCMIP, 6 models | +0.7 (+0.5 to +1.0) |
| | | *This study* | ECLIPSE, 3 models | +0.5 (+0.4 to +0.7) |
| $NO_X$ | $CH_4$ | Stevenson et al. (2013) | ACCMIP, 6 models | –5.5 (–7.4 to –4.2) |
| | | *This study* | ECLIPSE, 3 models, includes primary-mode $O_3$ | –0.4 to –2.1 (–2.6 to –2.5) depending on region |
| | $O_3$ | Stevenson et al. (2013) | ACCMIP, 6 models | +1.9 (+1.7 to +3.3) |
| | | *This study* | ECLIPSE, 3 models | +0.1 to +1.4 (+0.1 to +1.5) depending on region |
| | Aerosols | *This study* | ECLIPSE, 3 models, ari+aci | –0.3 to –0.8 (–1.2 to +0.2) depending on region |
| CO | $CH_4$ | Stevenson et al. (2013) | ACCMIP, 6 models | +0.11 (+0.07 to +0.13) |
| | | *This study* | ECLIPSE, 3 models, includes primary-mode $O_3$ | +0.12 to +0.15 (+0.08 to +0.20) depending on region |





| | | | | |
|---|---|---|---|---|
| | O$_3$ | Stevenson et al. (2013) | ACCMIP, 6 models | +0.11 (+0.08 to 0.14) |
| | | *This study* | ECLIPSE, 3 models | +0.03 to +0.06 (+0.03 to +0.07) depending on region |
| | Aerosols | *This study* | ECLIPSE, 3 models, ari+aci | +0.02 to +0.05 (−0.01 to +0.12) depending on region |
| NMVOC | CH$_4$ | Stevenson et al. (2013) | ACCMIP, 6 models | +0.27 (+0.00 to +0.41) |
| | | *This study* | ECLIPSE, 3 models, includes primary-mode O$_3$ | +0.35 to +0.66 (+0.02 to +0.93) depending on region |
| | O$_3$ | Stevenson et al. (2013) | ACCMIP, 6 models | +0.34 (+0.21 to +0.39) |
| | | *This study* | ECLIPSE, 3 models | +0.63 to +1.15 (+0.31 to +1.48) depending on region |
| | Aerosols | *This study* | ECLIPSE, 3 models, ari+aci | −0.18 to −0.74 (−1.48 to +0.86) depending on region |



**Table 2.** *List of models participating in the ECLIPSE radiative forcing simulations. Models*
*are either general circulation models (GCM) or chemistry-transport models (CTM).*
*Resolution indicates the horizontal resolution, in degrees, and the number of vertical levels.*
*Crosses indicate which aerosol species are represented in each model, among sulphate (SO4),*
*black carbon (BC), organic carbon (OC), secondary organic aerosol (SOA), and nitrate*
*(NO3) aerosols. Chemistry indicates whether the model includes an interactive tropospheric*
*ozone chemistry scheme. Radiation indicates whether radiation calculations are done*
*interactively (online) or offline from monthly distributions. Note that ozone radiative forcing*
*calculations are done offline for all models.*

| Model | Type | Resolution | SO$_4$ | BC | OC | SOA | NO$_3$ | Chemistry | Radiation |
|-------|------|------------|--------|----|----|-----|--------|-----------|-----------|
| ECHAM6-HAM2 | GCM | 1.8°x1.8° L31 | X | X | X | | | | Online |
| HadGEM3-GLOMAP | GCM | 1.8°x1.2° L38 | X | X | X | X | | X | Online |
| NorESM1-M | GCM | 1.9°x2.5° L26 | X | X | X | X | | X | Online |
| OsloCTM2 | CTM | 2.8°x2.8° L60 | X | X | X | X | X | X | Offline |





**Table 3.** *List of simulations made to provide radiative forcing by regional and seasonal*
*perturbations, and size of the emission perturbation applied to the anthropogenic component*
*for the year 2008, in Tg yr⁻¹. For some ozone precursors, HadGEM3 also perturbed the*
*biomass-burning component so the size of its perturbation is given in bracket (H:) for species*
*and regions with strong biomass-burning sources. Emitted masses are in [C] for black and*
*organic carbon, and volatile organic compounds. They are in [$NO_2$] for NOx.*

| # | Perturbation applied | Emission perturbation (Tg yr⁻¹) | |
|---|---|---|---|
| | | May—Oct | Nov—Apr |
| 1 | None (control simulation) | | |
| 2 | $SO_2$ emissions reduced by 20% in Europe | −0.77 | −0.85 |
| 3 | $SO_2$ emissions reduced by 20% in East Asia | −3.14 | −3.35 |
| 4 | $SO_2$ emissions reduced by 20% outside Europe, East Asia, and shipping sector | −5.1 | −5.2 |
| 5 | BC emissions reduced by 20% in Europe | −0.03 | −0.05 |
| 6 | BC emissions reduced by 20% in East Asia | −0.11 | −0.18 |
| 7 | BC emissions reduced by 20% outside Europe, East Asia, and shipping sector | −0.35 | −0.36 |
| 8 | OC emissions reduced by 20% in Europe | −0.04 | −0.07 |
| 9 | OC emissions reduced by 20% in East Asia | −0.21 | −0.37 |
| 10 | OC emissions reduced by 20% outside Europe, East Asia, and shipping sector | −0.80 | −0.83 |
| 11 | $NH_3$ emissions reduced by 20% in Europe | −0.39 | −0.39 |
| 12 | $NH_3$ emissions reduced by 20% in East Asia | −1.37 | −1.35 |
| 13 | $NH_3$ emissions reduced by 20% outside Europe, East Asia, and shipping sector | −3.48 | −3.43 |





| 14 | NOx emissions reduced by 20% in Europe | −1.00 | −1.06 |
|---|---|---|---|
| 15 | NOx emissions reduced by 20% in East Asia | −2.03 | −2.11 |
| 16 | NOx emissions reduced by 20% outside Europe, East Asia, and shipping sector | −6.27 (H: −7.17) | −6.37 (H: −6.69) |
| 17 | VOC emissions reduced by 20% in Europe | −0.06 to −0.28 | −0.07 to −0.36 |
| 18 | VOC emissions reduced by 20% in East Asia | −0.15 to −0.55 | −0.19 to −0.84 |
| 19 | VOC emissions reduced by 20% outside Europe, East Asia, and shipping sector | −0.15 to −4.08 | −0.19 to −4.17 |
| 20 | CO emissions reduced by 20% in Europe | −2.43 | −3.09 |
| 21 | CO emissions reduced by 20% in East Asia | −12.82 (H: −12.91) | −16.99 (H: −17.58) |
| 22 | CO emissions reduced by 20% outside Europe, East Asia, and shipping sector | −35.65 (H: −64.39) | −35.10 (H: −51.40) |
| 23 | All species of the shipping sector reduced by 20% | See Table 4. | |
| 24 | $CH_4$ perturbations equivalent to global 20% emission reduction | See ΔE in Table 7. | |





**Table 4.** *Size of the emission perturbation applied to the shipping sector for the year 2008, in*
*Tg yr$^{-1}$. Emitted masses are in [C] for black and organic carbon, and volatile organic*
*compounds. They are in [NO$_2$] for NOx. Emissions used in ECHAM6 and NorESM1 are*
*denoted with E and N, where different.*

| Species | Emission perturbation (Tg yr$^{-1}$) | |
|---|---|---|
| | **May—Oct** | **Nov—Apr** |
| SO$_2$ | −1.04 (E: −1.25) | −1.04 (E: −1.24) |
| BC | −0.01 (E: −0.02) | −0.01 (E: −0.02) |
| OC | −0.01 (E: −0.02) | −0.01 (E: −0.02) |
| NO$_X$ | −1.70 (N: −1.10) | −1.67 (N: −1.10) |
| VOC | −0.04 to −0.21 | −0.04 to −0.21 |
| CO | −0.11 | −0.11 |



1    **Table 5**. *Simulated lifetime, in days, of aerosol species and tropospheric ozone in the four*

2    *participating models.*

| Species | ECHAM6 | HadGEM3 | NorESM1 | OsloCTM2 |
|---|---|---|---|---|
| Sulphate | 4.0 | 5.2 | 4.2 | 3.5 |
| BC | 5.2 | 5.7 | 8.0 | 6.2 |
| OC | 5.0 | 6.6 | 7.7 | 5.0 |
| Tropospheric ozone | n/a | 20.7 | 26.4 | Not diagnosed |



**Table 6.** *Semi-direct radiative forcing (SDRF) by regional and seasonal perturbations of*
*black carbon aerosols. Column 3 gives the scaling factor imposed to let rapid adjustments*
*from the semi-direct effect emerge from natural variability. Column 4 gives the corresponding*
*specific SDRF, in mW m$^{-2}$ (Tg[C] yr$^{-1}$)$^{-1}$, and its standard deviation over the 30 years.*

| Region | Season | Scaling factor | Specific SDRF |
|---|---|---|---|
| Europe | Summer | 500 | -31 ± 13 |
| | Winter | 500 | -3 ± 8 |
| East Asia | Summer | 150 | -38 ± 12 |
| | Winter | 150 | +1 ± 7 |
| Global | Summer | 30 | -40 ± 18 |
| | Winter | 30 | -14 ± 11 |



**Table 7**. *Characteristics of the methane budget in ECLIPSE models. For NorESM1, numbers*
*are given for the Summer perturbation simulation. From left to right, columns give: methane*
*lifetime to destruction by OH, $\tau_{OH}$ in years, for the control (Ctl) and perturbed (Per)*
*simulations; total methane lifetime, $\tau_{tot}$ in years, in Ctl and Per simulations; total methane*
*burden, B in Tg[CH$_4$], in Ctl and Per simulations; methane feedback factor f; equivalent*
*methane emission perturbation, $\Delta E$ in Tg[CH$_4$] yr$^{-1}$; methane radiative forcing, RF in*
*mW m$^{-2}$; methane specific radiative forcing, SRF in mW m$^{-2}$ (Tg[CH$_4$] yr$^{-1}$)$^{-1}$. See Sect. 3.5 for*
*details.*

| Model | $\tau_{OH}$ | | $\tau_{tot}$ | | B | | $f$ | $\Delta E$ | RF | SRF |
|---|---|---|---|---|---|---|---|---|---|---|
| | Ctl | Per | Ctl | Per | Ctl | Per | | | | |
| HadGEM3 | 6.0 | 5.6 | 5.5 | 5.2 | 4561 | 3702 | 1.34 | 117 | 123 | 1.21 |
| NorESM1 | 7.8 | 7.7 | 7.0 | 6.9 | 4815 | 4489 | 1.28 | 36.5 | 44 | 1.38 |
| OsloCTM2 | 10.2 | 9.6 | 8.9 | 8.4 | 4909 | 4115 | 1.46 | 61 | 109 | 2.04 |



**7   Figures**
**Figure 1**. *Simplified description of tropospheric ozone chemistry. Arrows represent chemical*
*reactions and, for processes noted in italics, sources and sinks. For the sake of simplicity, the*
*role of volatile organic compounds is not shown, but discussed in the main text.*

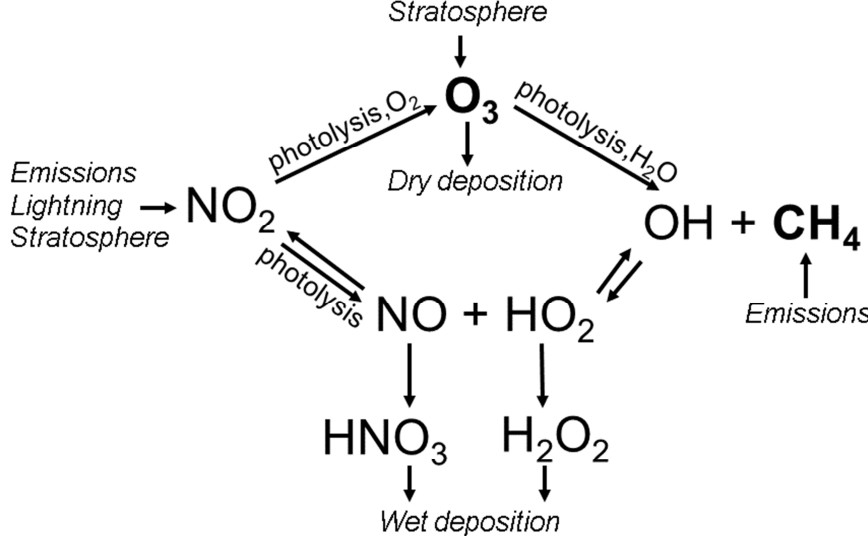




1   **Figure 2.** *HTAP tier-1 regions used in the ECLIPSE specific radiative forcing matrix. EU*

2   *stands for Europe and EA for East Asia.*

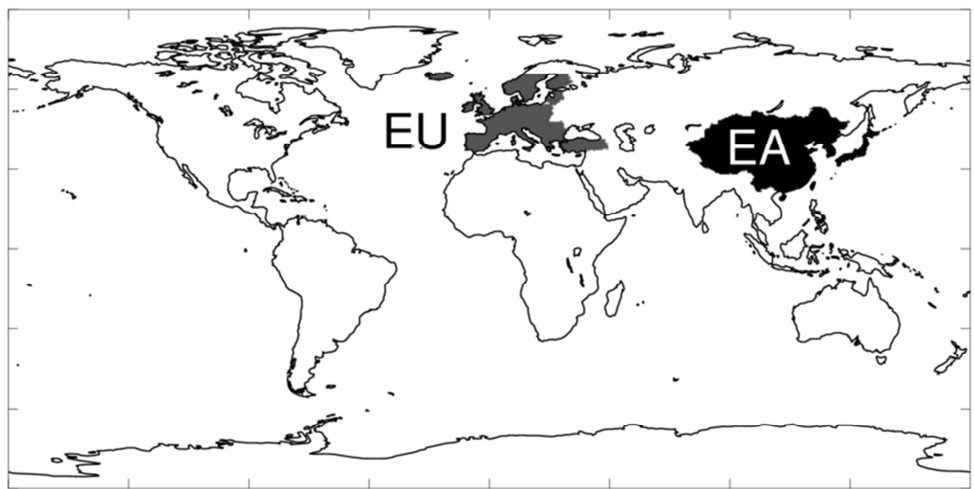



**Figure 3.** *Specific radiative forcing, in mW m$^{-2}$ (Tg[SO$_2$] yr$^{-1}$)$^{-1}$, for regional and seasonal*
*reductions in sulphur dioxide emissions. Results are obtained by four global models:*
*OsloCTM2 (O), NorESM1 (N), HadGEM3 (H), and ECHAM6 (E). Radiative forcing is*
*diagnosed as the sum of aerosol-radiation and aerosol-cloud interactions, except for*
*ECHAM6 which diagnoses aerosol-radiation interactions only.*

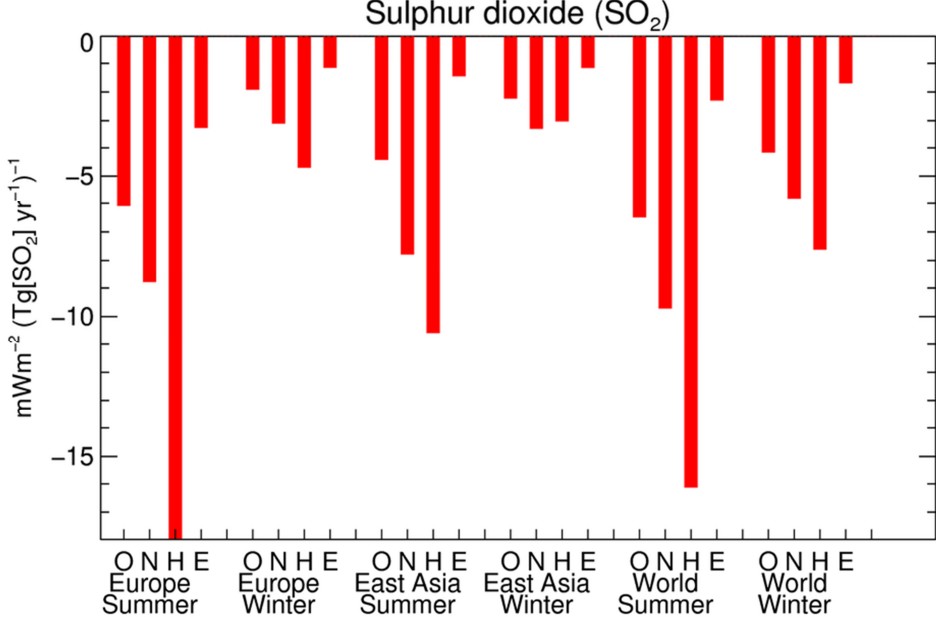





**Figure 4.** *Specific radiative forcing, in mW m⁻² (Tg[C] yr⁻¹)⁻¹, for regional and seasonal*
*reductions in primary black carbon aerosol emissions. Results are obtained by four global*
*models: OsloCTM2 (O), NorESM1 (N), HadGEM3 (H), and ECHAM6 (E). Three categories*
*of radiative forcing mechanisms are included: aerosol-radiation and aerosol-cloud*
*interactions (red, except for ECHAM6 where aerosol-cloud radiative forcing is not*
*diagnosed), BC deposition on snow (grey, OsloCTM2 only), and rapid adjustments from the*
*semi-direct effect of BC (blue, OsloCTM2 only).*

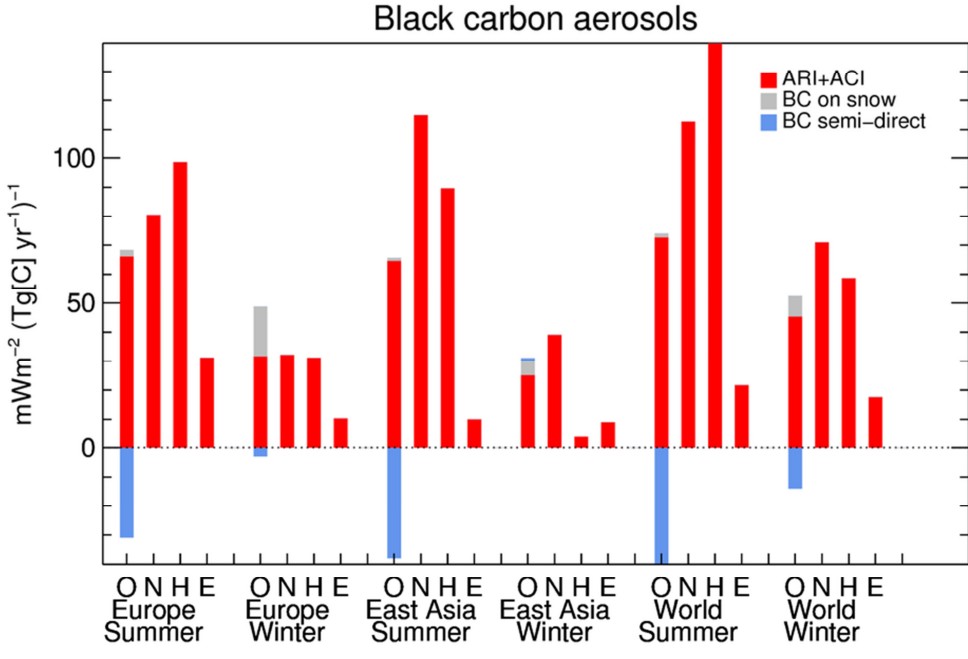



**Figure 5.** *Specific radiative forcing, in mW m$^{-2}$ (Tg[C] yr$^{-1}$)$^{-1}$, for regional and seasonal*
*reductions in primary organic carbon aerosol emissions. Results are obtained by four global*
*models: OsloCTM2 (O), NorESM1 (N), HadGEM3 (H), and ECHAM6 (E). Radiative forcing*
*is diagnosed as the sum of aerosol-radiation and aerosol-cloud interactions, except for*
*ECHAM6 which diagnoses aerosol-radiation interactions only.*

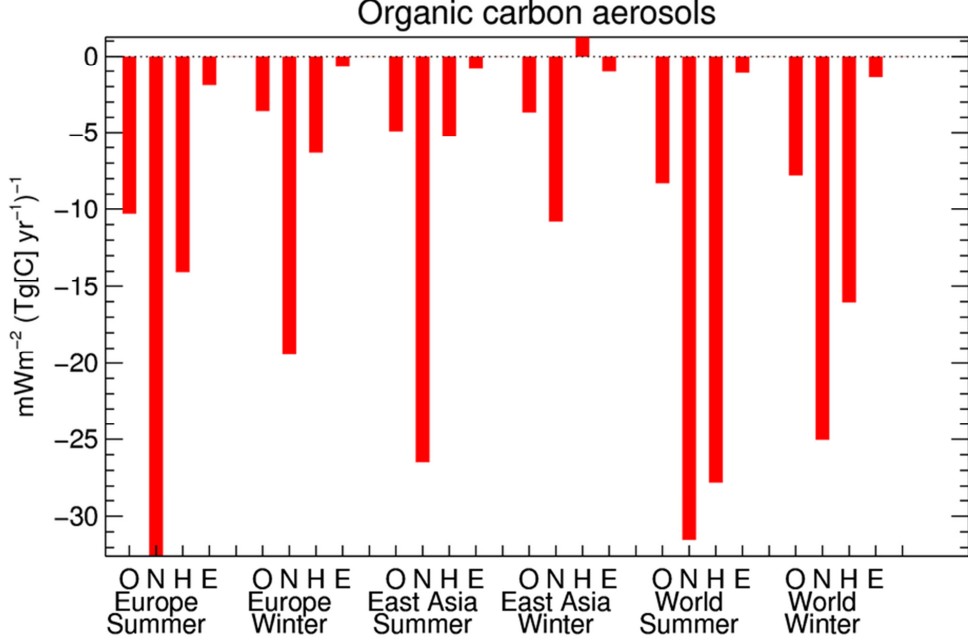





1    **Figure 6.** *Specific radiative forcing, in mW m⁻² (Tg[NH₃] yr⁻¹)⁻¹, for regional and seasonal*

2    *reductions in ammonia emissions. Results are obtained by the OsloCTM2 model, and include*

3    *aerosol-radiation and aerosol-cloud interactions.*

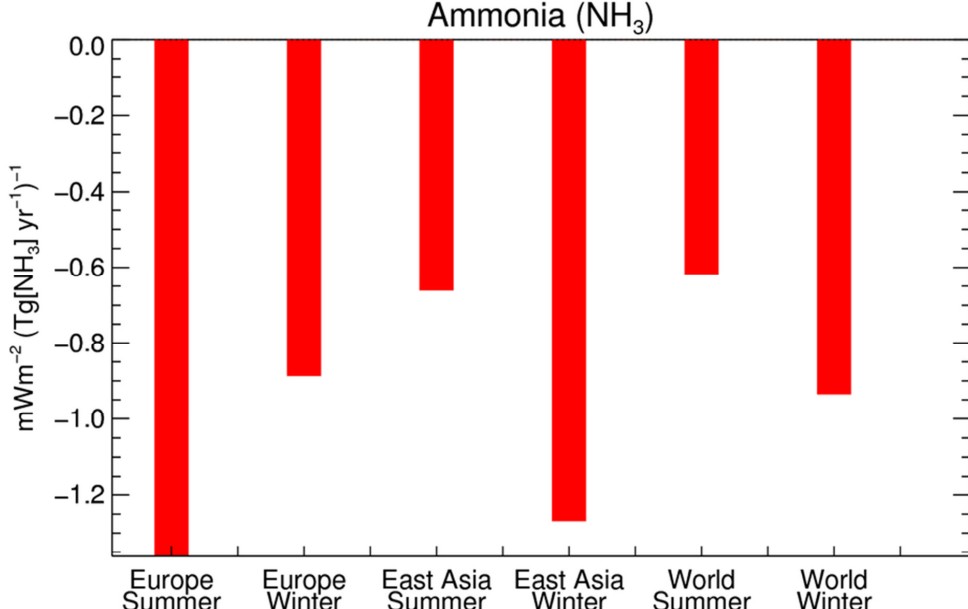





**Figure 7.** *Specific radiative forcing, in mW m$^{-2}$ (Tg[CH$_4$] yr$^{-1}$)$^{-1}$, for global and annual*
*reductions in equivalent methane emissions (see Sect. 3.5 for details). Results are obtained by*
*three global models: OsloCTM2 (O), NorESM1 (N), and HadGEM3 (H). Three categories of*
*radiative forcing mechanisms are included: aerosol-radiation and aerosol-cloud interactions*
*(red), short-term changes in ozone (blue), and methane (yellow).*

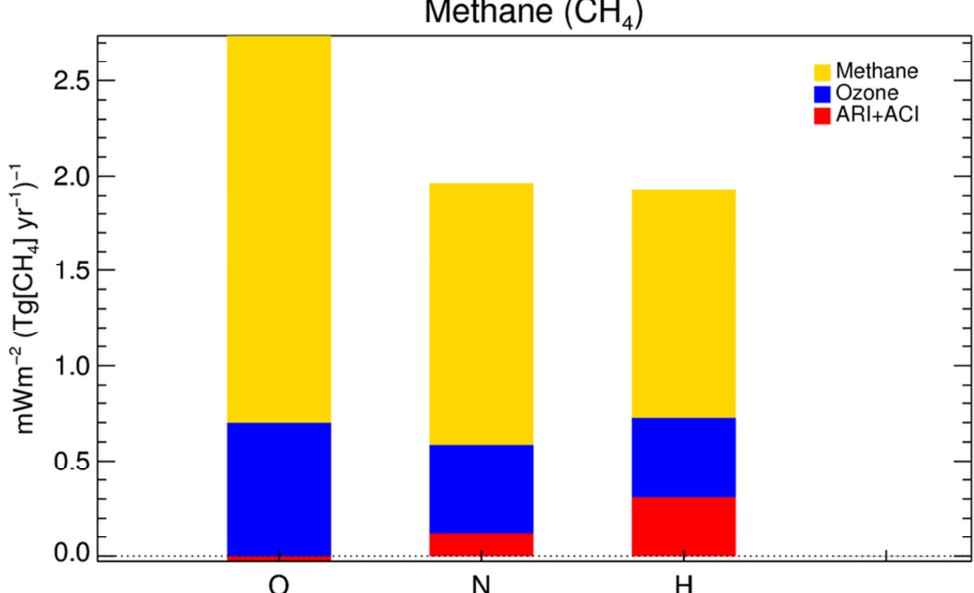





**Figure 8.** *Specific radiative forcing, in mW m$^{-2}$ (Tg[NO$_2$] yr$^{-1}$)$^{-1}$, for regional and seasonal*
*reductions in nitrogen oxide emissions. Results are obtained by three global models:*
*OsloCTM2 (O), NorESM1 (N), and HadGEM3 (H). Four categories of radiative forcing*
*mechanisms are included: aerosol-radiation and aerosol-cloud interactions (red), short-term*
*changes in ozone (blue), methane (yellow), and primary-mode ozone (green).*

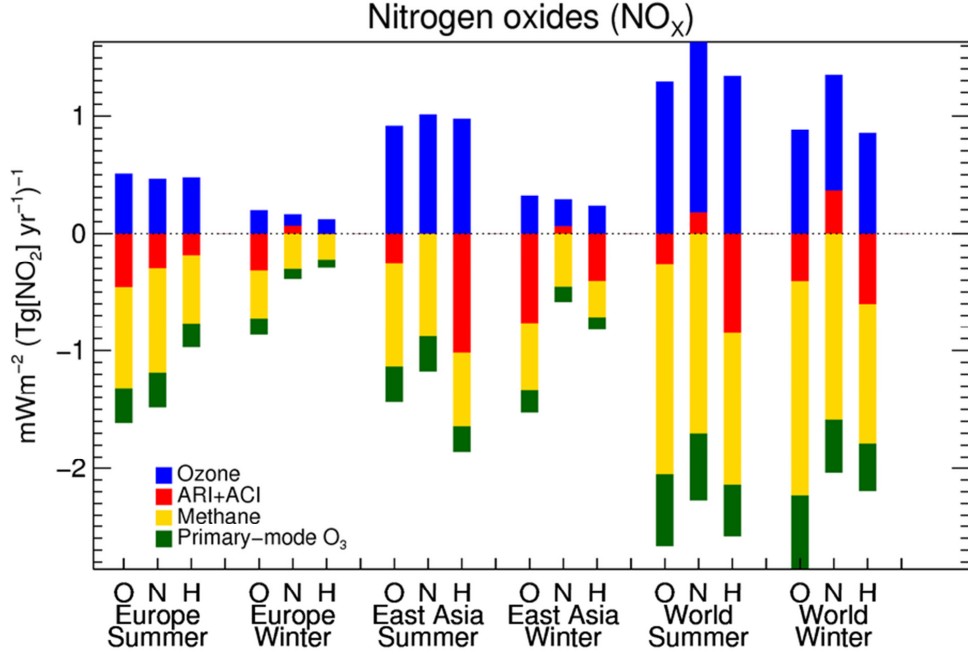





**Figure 9.** *Specific radiative forcing, in mW m$^{-2}$ (Tg[C] yr$^{-1}$)$^{-1}$, for regional and seasonal*
*reductions in emissions of volatile organic compounds. Results are obtained by three global*
*models: OsloCTM2 (O), NorESM1 (N), and HadGEM3 (H). Four categories of radiative*
*forcing mechanisms are included: aerosol-radiation and aerosol-cloud interactions (red),*
*short-term changes in ozone (blue), methane (yellow), and primary-mode ozone (green).*

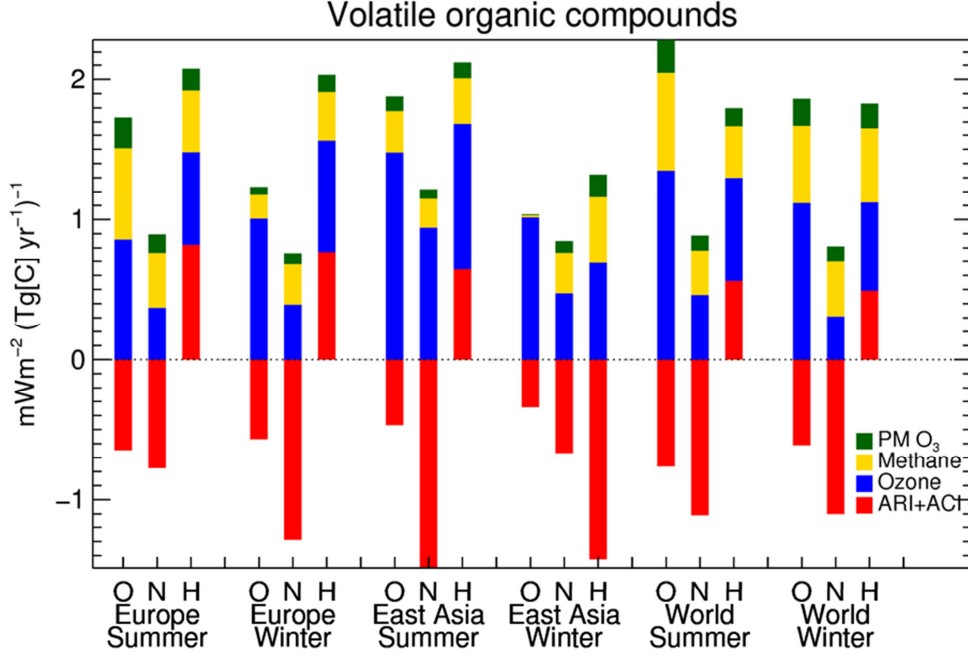





**Figure 10.** *Specific radiative forcing, in mW m$^{-2}$ (Tg[CO] yr$^{-1}$)$^{-1}$, for regional and seasonal*
*reductions in emissions of carbon monoxide. Results are obtained by three global models:*
*OsloCTM2 (O), NorESM1 (N), and HadGEM3 (H). Four categories of radiative forcing*
*mechanisms are included: aerosol-radiation and aerosol-cloud interactions (red), short-term*
*changes in ozone (blue), methane (yellow), and primary-mode ozone (green).*

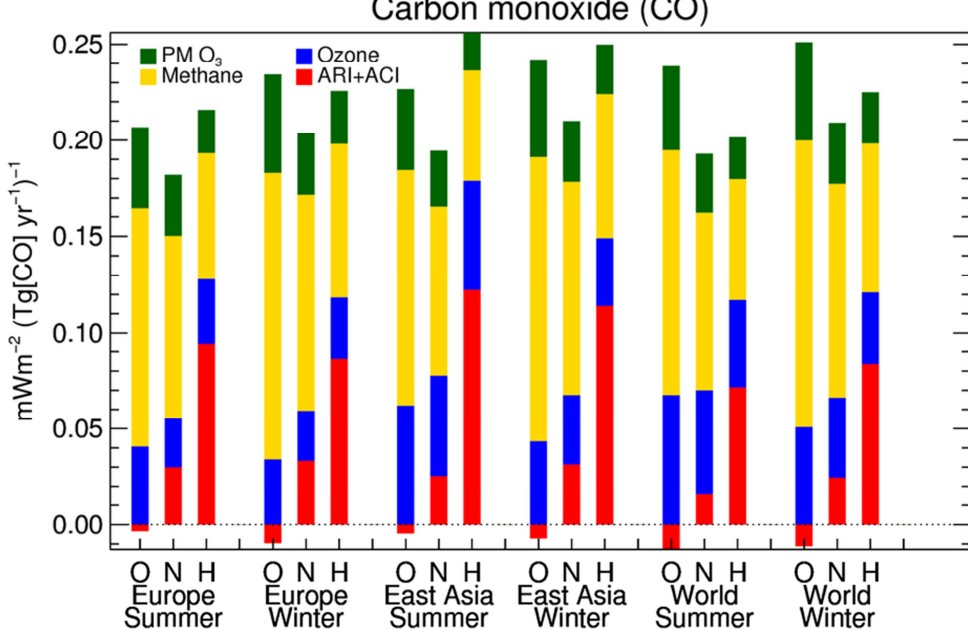



**Figure 11.** *Specific radiative forcing, in mW m$^{-2}$ (Tg yr$^{-1}$)$^{-1}$, for seasonal reductions in all the*
*species emitted by the shipping sector. The species included and their units of emitted mass*
*are sulphur dioxide (SO$_2$), black carbon (C), organic carbon (C), ammonia (NH$_3$), nitrogen*
*oxides (NO$_2$), volatile organic compounds (C), carbon monoxide (CO), and methane (CH$_4$).*
*Results are obtained by four global models: OsloCTM2 (O), NorESM1 (N), HadGEM3 (H),*
*and ECHAM6 (E). Six categories of radiative forcing mechanisms are included: aerosol-*
*radiation and aerosol-cloud interactions (red, except for ECHAM6 which diagnoses aerosol-*
*radiation only), black carbon deposition on snow (grey, OsloCTM2 only), black carbon rapid*
*adjustments from the semi-direct effect (light blue, OsloCTM2 only), short-term changes in*
*ozone (dark blue, not simulated by ECHAM6), methane (yellow, not simulated by ECHAM6),*
*and primary-mode ozone (green, not simulated by ECHAM6).*

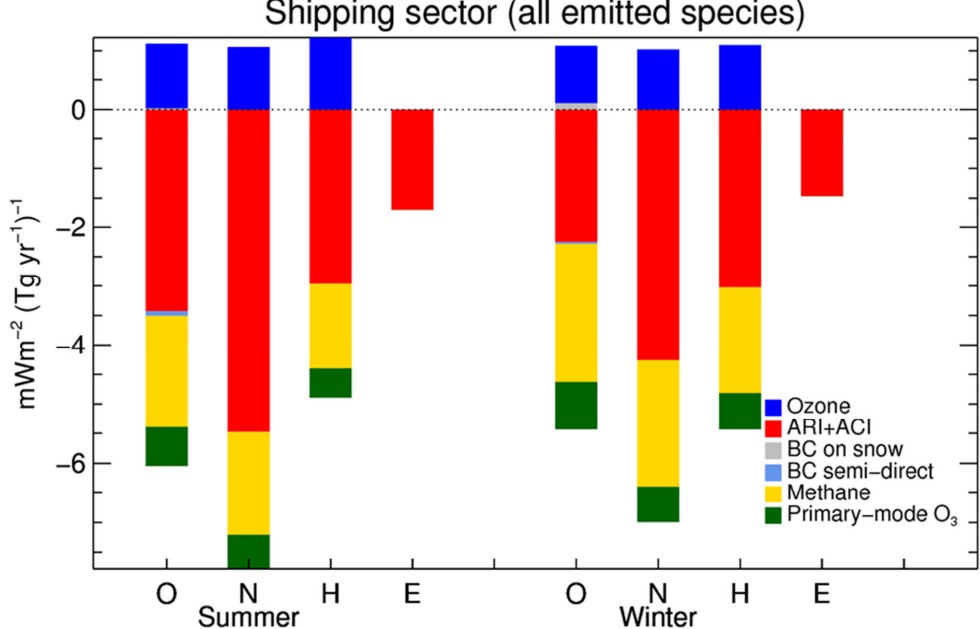




**Figure 12.** *Best estimates of specific radiative forcing for regional and seasonal reductions in*
*near-term climate forcer emissions, in mW m$^{-2}$ (Tg yr$^{-1}$)$^{-1}$. Best estimates are given for six*
*categories of radiative forcing: aerosol-radiation and aerosol-cloud interactions (red), black*
*carbon deposition on snow (grey), black carbon rapid adjustments from semi-direct effects*
*(light blue), short-term changes in ozone (dark blue), methane (yellow), and primary-mode*
*ozone (green). Black bars show the total specific radiative forcing, i.e. the sum of the six*
*components listed above, and whiskers denote the weakest and strongest specific radiative*
*forcing that are obtained by the four participating models or, in the case of ammonia*
*perturbations, estimated from the literature.*

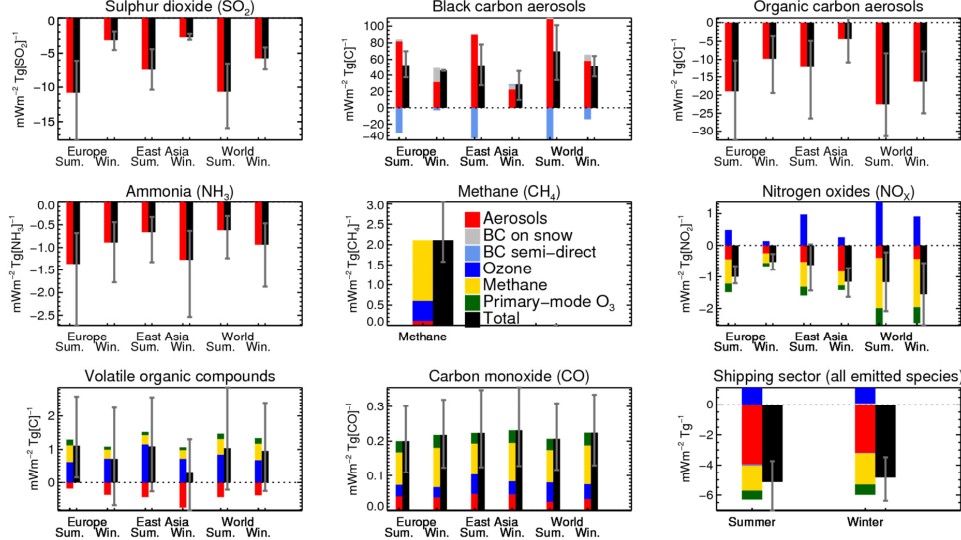





**Figure 13a**. *Best estimates of annually-averaged specific radiative forcing, in mW m$^{-2}$*
*(Tg[species] yr$^{-1}$)$^{-1}$, in four latitude bands, for aerosol primary and precursor emission*
*perturbations. Each row corresponds to a perturbed species: from top to bottom, sulphur*
*dioxide, black carbon, organic carbon, and ammonia. Each column corresponds to a regional*
*and seasonal perturbation. Barcharts are shown for four latitude bands, from left to right:*
*90N—60N, 60N—28N, 28N—28S, and 28S—90S.*



1  **Figure 13b.** *As Figure 13a, but for ozone precursor and shipping sector perturbations.*
2  *Perturbed species are, from top to bottom, nitrogen oxide, volatile organic compounds,*
3  *carbon monoxide, methane, and all species emitted by the shipping sector.*

