# Peer review of "Regional and seasonal radiative forcing by perturbations to"

_Atmospheric Chemistry and Physics, 2016_

## Referee Comment (RC1) · Anonymous Referee #1 · 1 Jul 2016

The manuscript documents results of specific radiative forcing (SRF) for near-term climate forcers (NTCFs) from multi-model simulations conducted under the auspices of the European project - Evaluating the Climate and Air Quality Impacts of Short-lived Pollutants (ECLIPSE). SRFs for NTCF emissions from two source regions, the shipping sector and global contributions from four global models (three chemistry-climate models and one chemical transport model) are discussed in detail providing some explanation of diversity in the results. The analysis generates estimates of region- and sector-specific SRFs and shows that diversity in these estimates comes from differences in the configuration of the control simulations in addition to the structural differences (e.g. not all models explicitly representation of aerosol indirect effects). The

paper falls within the scope and aims of ACP and is appropriate for publication. However, I found it difficult to discern new scientific knowledge generated in this work that will help advance our understanding of the influence of NTCF on climate. As such, SRF is just another way or metric for evaluating a species' radiative forcing (RF). I understand the need for considering RFs in the context of emissions of NTCFs (or their precursors), but the emissions; even the present-day estimates, themselves are highly uncertain (e.g., Granier et al., 2011). Further, from past research (as highlighted on page 5), we know that BC aerosols, methane and carbon monoxides exert positive RF, and OC, SO2, and NH3 exert negative RF. How important are monthly/seasonal RFs for understanding surface temperature response to NTCFs? If the intention of the paper is to motivate discussions on including NTCFs in a climate mitigation policy, I am not sure what to make of such diverse SRF estimates. Analyzing multi-model results is no mean feat especially in the face of model structural diversity, inconsistent simulations, and incomplete output. I commend the authors for their efforts, however, I question the value of such diverse numbers produced in this manuscript. I am, therefore, unable to recommend the publication of this paper in its current form. Below, I provide some specific comments to address issues with the paper.

Specifica Comments:

The introduction is too long. It starts out by providing a text-book summary of the radiative influence of NTCFs, describes tropospheric ozone chemistry and interactions that have radiative feedbacks, and so on until the last paragraph of page 7 where the aims of the study is described. I think much of the information up until page 7 can be condensed. For example, the tropospheric ozone chemistry and its interactions with aerosols has been covered extensively in several review papers - some recent ones are Schneidemesser et al., (2015) and Fiore et al., (2015). Paragraph 1 on page 3 and para 2 on page 4 can be combined to define RF/ERF calculations and describe the influence of aerosols. Finally, the focus should be on why regional and seasonal SRFs are important, what do we know about NTCF SRFs from previous studies (2nd paragraph

on page 5) and how does this study advance the knowledge base by systematically analyzing SRFs from different models.

Page 4, line 9: Typo 'Goddart'

Page 9, lines 3-5: Please give references here.

Models and Experiment protocols: It is not clear if the 1 year simulations of CCMs (ECHAM6-HAM2, HADGEM3-GLOMAP and NorESM1) are performed in free-running mode or with fixed SST and sea-ice, or in the nudged mode with meteorological fields from reanalysis.

Page 10 lines 20-21: It would be helpful to see the spatial distribution of NTCF (or their precursor) emissions.

Page 10, Lines 21-22: It is not entirely clear what "heating" the authors are talking about. One can guess that this is the winter time heating of homes, but then the audience should not have to guess, right? Is there any analysis of how realistic the seasonal cycle in domestic emissions is?

Page 10, line 29-30: It is mentioned that results of "that last region" (Rest of the World-RotW) are obtained by adding Europe, East Asia and RotW. Why are all of these added to produce Rest of the World SRFs?

Page 11, line 1: Reference needed after "climate policy objectives".

Page 11, line 7: Referenced needed after "policy agenda".

Page 11, lines 15-17: Is it too late to correct these mistakes? These obviously add to the diversity in the SRFs for shipping.

Page 11, lines 17-21: Differences in the VOC species considered by the models (because of differences in chemical mechanisms implemented in the model) are a significant source of diversity in the simulated tropospheric ozone (Young et al., 2013). This should be highlighted here.

Page 11, lines 23-25: Do the models prescribe global mean surface CH4 concentrations or apply a latitudinal variation?

Page 13, 1st paragraph: How do methane lifetimes compare across models? ACCMIP studies have revealed diversity in simulated OH and CH4 lifetimes (Naik et al., 2013; Voulgarakis et al., 2013). Do the models considered here also suffer from this diversity?

Page 13, line 3: "BC lifetime is longer" than what? The statement is not clear.

Page 13, lines 7-10: This is a very broad-brush way of dealing with diversity. Can we not learn anything about what drives diversity in lifetimes from further analysis of model output?

Page 13, lines 18-22: Please qualify these statements with references or point the reader to a particular table that lists the papers that evaluated ECLIPSE models.

Biases and scaling of specific radiative forcing: Please consider condensing this section. There is no new evaluation presented in the two paragraphs on page 14. The discussion is referring to results from other papers, which can be succinctly summarized in section 3 to explain the results.

Section 3 - How do the SRFs for O3 and its precursors calculated here compare with estimates from previous studies including - Naik et al. (2005), Fry et al., (2012), Fry et al. (2013). Fry et al. (2014)

Page 16, lines 21-23, 28-30: I am not sure if I follow the discussion here. On lines 21-22, it is mentioned that the ari is stronger in winter than in summer but then on line it is mentioned that the SO2 is SRF is stronger for summer than winter. This appears contradictory.

Page 18, lines 15-17: Please insert a reference here to support this statement that O3 and CH4 RF are affected by BC perturbations via heterogeneous reactions.

Page 22, section 3.3: Are secondary organic aerosols (SOA) included in the quantification of OC SRFs?

Page 38, lines 19-21: There are several studies in the literature that have studied regional O3 SRFs some of which have been highlighted in my comments above. These need to acknowledged here.

References: Granier et al., (2011), Evolution of anthropogenic and biomass burning emission of air pollutants at global and regional scales during the 1980–2010 period, Clim. Change,109, 163–190.

Fiore et al., (2015), Air quality and climate connections, J. of Air & Waste Management, 65:6, 645-685, doi: 10.1080/10962247.2015.1040526.

Fry et al. (2012), The influence of ozone precursor emissions from four world regions on tropospheric composition and radiative climate forcing, J. Geophy. Res., 117, D07306, DOI:10.1029/2011JD017134

Fry et al. (2013), Net radiative forcing and air quality responses to regional CO emission reductions, Atmos. Chem. Phys., 13, 5381-5399, doi:10.5194/acp-13-5381-2013.

Fry et al (2014), Air quality and radiative forcing impacts of anthropogenic volatile organic compound emissions from ten world regions, Atmos. Chem. Phys., 14, 523–535, 2014

Naik et al. (2005), Net radiative forcing due to changes in regional emissions of tropospheric ozone precursors, J. Geophys. Res., 110, doi:10.1029/2005JD005908

Naik et al. (2013), Preindustrial to present day changes in tropospheric hydroxyl radical and methane lifetime from the Atmospheric Chemistry and Climate Model Intercomparison Project (ACCMIP), Atmos. Chem. Phys., 13, 5277-5298, doi:10.5194/acp-13-5277-2013

von Schneidemesser et al., (2015), Chemistry and the Linkages between Air Quality and Climate Change, Chem. Rev. 2015, 115, 3856−3897, DOI:

10.1021/acs.chemrev.5b00089.

Voulgarakis et al. (2013), Analysis of present day and future OH and methane lifetime in the ACCMIP simulations, Atmos. Chem. Phys., 13, 2563-2587, doi:10.5194/acp-13-2563-2013

Young et al. (2013), Pre-industrial to end 21st century projections of tropospheric ozone from the Atmospheric Chemistry and Climate Model Intercomparison Project (ACCMIP), Atmos. Chem. Phys., 13, 2063-2090.

---

## Referee Comment (RC2) · Anonymous Referee #2 · 11 Jul 2016

This paper documents the results of a collection of experiments aimed at identifying the forcing associated with regional perturbations of emissions. It does so using a collection of 4 models, with varying degrees of complexity on chemistry and aerosol representation, amongst other sources of differences. This is a timely paper, focusing on policy-relevant questions.

Main comment

While I find the scientific approach interesting and worthwhile, I find the paper in this present form to be quite limited. It reads mostly as a report, with considerable discussion of all findings, but little added understanding or combined pieces of information. More specifically, the paper is too long and should focus much more on summaries

(such Figure 12) than the description of each aspect and model results separately. This is because models are different (see for example the discussion on page 13, lines 1-15) and so give different forcings, but the paper only describes differences (as it is explicitly noted on Page 15, lines 30 and 31 and Page 16 lines 1-3). I think that the authors should focus on the important information that comes out of their analysis, and put all the more detailed discussion into the supplement. This will make the paper much more readable and useful, the way Shindell and Faluvegi has been.

Minor comments

Page 5, lines 22-25: since several forcers have negative forcings, it would be better to list the ones with the positive forcings instead of using "All . . ." Page 12, lines 13-19: this definition would not be sufficient for computing RFaci (see Ghan, 2013; http://www.atmos-chem-phys.net/13/9971/2013) Page 23, lines 26-27: this kind of comment has to be substantiated. Similarly, Page 25, lines 16-18.
* * *

---

## Author Comment (AC1) · 30 Sep 2016

**Answers to reviews of "Regional and seasonal radiative forcing by perturbations to aerosol and ozone precursor emissions" by Bellouin et al., submitted to *Atmos. Chem. Phys.***

We thank the two anonymous reviewers for their useful reviews of the manuscript and the relevant references they provided. Those reviews have led to major revisions, which we think have much improved the manuscript. Those revisions include:

- A shortening of the manuscript, as recommend by both reviewers. 7 pages of text and 6 figures have been removed.
- The motivation for the study is not clearer, the original findings are better highlighted, and the implications for climate mitigation decision-making are more clearly discussed.

We however chose to retain some emphasis on the discussion of model diversity. That choice comes as a compromise to best serve the two audiences interested in the paper: users of the radiative forcing matrix, who need to be assured that it captures model spread and identifies the more uncertain aspects, and atmospheric composition modellers, who need to know where model diversity comes from in order to eventually reduce it.

In addition, a mistake in Table 1, where black carbon specific radiative forcings from Bond et al. (2013) were wrong, has been corrected.

In the following, reviewer comments are in italics and excerpts of the revised manuscript in bold.

Response to main comments:

**Reviewer 1:** […] *The paper falls within the scope and aims of ACP and is appropriate for publication. However, I found it difficult to discern new scientific knowledge generated in this work that will help advance our understanding of the influence of NTCF on climate. As such, SRF is just another way or metric for evaluating a species' radiative forcing (RF). I understand the need for considering RFs in the context of emissions of NTCFs (or their precursors), but the emissions; even the present-day estimates, themselves are highly uncertain (e.g., Granier et al., 2011). Further, from past research (as highlighted on page 5), we know that BC aerosols, methane and carbon monoxides exert positive RF, and OC, SO2, and NH3 exert negative RF. How important are monthly/seasonal RFs for understanding surface temperature response to NTCFs? If the intention of the paper is to motivate discussions on including NTCFs in a climate mitigation policy, I am not sure what to make of such diverse SRF estimates. Analyzing multi-model results is no mean feat especially in the face of model structural diversity, inconsistent simulations, and incomplete output. I commend the authors for*

*their efforts, however, I question the value of such diverse numbers produced in this manuscript. I am, therefore, unable to recommend the publication of this paper in its current form.*

**Answer:** We agree with the reviewer that novel findings were hidden in the first version of the manuscript. The abstract, the introduction, and the conclusion sections have been rewritten to highlight the novelty and potential impacts of the study.

The novel findings we now draw attention to are:
- An upward revision of sulphur dioxide and organic carbon specific radiative forcings because aerosol-cloud interactions are now included. In contrast, there is a lack of agreement on the sign of the specific radiative forcing of volatile organic compound perturbations, so basing mitigation policies on those seems unwise;
- The strong seasonalities of the specific radiative forcing of most forcers. Playing on that seasonality allows strategies to minimise positive radiative forcing based on the timing of emissions. For example, if the aim is to improve air quality while minimising the reduction in negative aerosol radiative forcing that comes from reduced emissions, then privileging wintertime emission reductions seems a good choice, because radiative forcing is smaller then.
- The stronger aerosol specific radiative forcings exerted by European emissions compared to East Asia where the baseline is more polluted. Again, if the aim is to improve air quality without incurring a "climate penalty", then reductions in the more polluted regions should come first, because the radiative forcing is saturated there so will not change much when emissions are reduced.

The reviewer is also disappointed by the large diversity in the estimates that we obtain from the four models. The difficulty to obtain a consistent set of simulations and diagnostics is indeed disappointing, but that is the challenge faced by all multi-model studies, including AeroCom or CMIP. Model structural diversity is not going to go away soon, because different climate modelling centres have different resources and priorities for model development. It is therefore important to capture that diversity, and to ensure that users of multi-model datasets like ours are aware of the extent and causes of the diversity. This is what our study proposes to do, and why we are keen to preserve the in-depth discussion on model diversity.

**Reviewer 2:** *This paper documents the results of a collection of experiments aimed at identifying the forcing associated with regional perturbations of emissions. It does so using a collection of 4 models, with varying degrees of complexity on chemistry and aerosol representation, amongst other sources of differences. This is a timely paper, focusing on policy-relevant questions. While I find the*

*scientific approach interesting and worthwhile, I find the paper in this present form to be quite limited. It reads mostly as a report, with considerable discussion of all findings, but little added understanding or combined pieces of information. More specifically, the paper is too long and should focus much more on summaries (such Figure 12) than the description of each aspect and model results separately. This is because models are different (see for example the discussion on page 13, lines 1-15) and so give different forcings, but the paper only describes differences (as it is explicitly noted on Page 15, lines 30 and 31 and Page 16 lines 1-3). I think that the authors should focus on the important information that comes out of their analysis, and put all the more detailed discussion into the supplement. This will make the paper much more readable and useful, the way Shindell and Faluvegi has been.*

**Answer**: We thank the reviewer for the positive comments, and agree with the criticism that the first version of the manuscript was not as good as it should have been. Reaching the level of interest of Shindell and Faluvegi (2009) would be excellent and we have therefore rewritten the abstract and main text to better emphasise the important information that comes from the radiative forcing matrix. See the response to reviewer 1 above for the three key messages of the paper that are now highlighted in the abstract and conclusion sections. We believe those changes, added to the shortening of the manuscript in general, made the paper more readable and useful, as anticipated by the reviewer.

We however think that model diversity deserves emphasis, for two reasons. First, the users of the radiative forcing matrix (which is the audience interested in Shindell and Faluvegi) will want to know that model diversity is captured and its causes understood. Second, climate modellers (which may not have been the primary target audience of Shindell and Faluvegi) will want to refer to the paper to assess development priorities and diversity changes.

Response to specific comments:

**Reviewer 1:** *The introduction is too long. It starts out by providing a text-book summary of the radiative influence of NTCFs, describes tropospheric ozone chemistry and interactions that have radiative feedbacks, and so on until the last paragraph of page 7 where the aims of the study is described. I think much of the information up until page 7 can be condensed. For example, the tropospheric ozone chemistry and its interactions with aerosols has been covered extensively in several review papers - some recent ones are Schneidemesser et al., (2015) and Fiore et al., (2015). Paragraph 1 on page 3 and para 2 on page 4 can be combined to define RF/ERF calculations and describe the influence of aerosols. Finally, the focus should be on why regional and seasonal SRFs are important, what do we know about NTCF SRFs from previous studies (2nd paragraph on page 5) and how does this study advance the knowledge base by systematically analyzing SRFs from different models.*

**Answer:** Thanks to the reviewer's comments, we have taken the time to write a shorter introduction, which has been reduced by a third compared to the first version of the manuscript. We now refer the reader to von Schneidemesser et al. (2015) and Fiore et al. (2015) for details of the intricacies of atmospheric ozone and aerosol interactions.

The importance of accounting for all main radiative forcing mechanisms and for the regional and seasonal character of near-time climate forcer emissions is also now discussed in the introduction:

**[…] several policy choices are not addressed by existing studies. First, they do not include all radiative forcing mechanisms consistently. RFaci and contributions to BC RF from deposition on snow and rapid adjustments from the semi-direct effect are often excluded. Then, although it is clearly important to take a regional view like that of Fry et al. (2012) and Yu et al. (2013), it is potentially equally important to account for the seasonality of the emissions. RF mechanisms based on perturbations of sunlight are obviously strongly seasonal, so it is misleading to use year-long perturbations to quantify mitigation options that mostly act, because of the short lifetimes of NTCFs, for wintertime (e.g. domestic heating) or summertime (e.g. air conditioning) periods.**
**To remove those limitations, the Evaluating the CLimate and Air Quality ImPacts of Short-livEd Pollutants (ECLIPSE) project (Stohl et al., 2015) built a matrix of SRFs that includes several NTCFs, varies the region and time of emissions, and spans diversity among models.**

**Reviewer 1:** *Page 4, line 9: Typo 'Goddart'*
**Answer:** Thank you but that part of the statement has been removed during revisions.

**Reviewer 1:** *Page 9, lines 3-5: Please give references here.*
**Answer:** We now cite Myhre et al. (2013b), Stevenson et al. (2013) and Shindell et al. (2013) to support the statement that ECLIPSE models are representative of total model diversity.

**Reviewer 1:** *Models and Experiment protocols: It is not clear if the 1 year simulations of CCMs (ECHAM6-HAM2, HADGEM3-GLOMAP and NorESM1) are performed in free-running mode or with fixed SST and sea-ice, or in the nudged mode with meteorological fields from reanalysis.*
**Answer:** We have clarified that all simulations are free-running with fixed SST and sea-ice.

**Reviewer 1:** Page 10 lines 20-21: It would be helpful to see the spatial distribution of NTCF (or their precursor) emissions.
**Answer:** We have chosen instead to point the reader to the location where the dataset is stored: http://www.iiasa.ac.at/web/home/research/researchPrograms/air/ECLIPSEv4a.html

**Reviewer 1:** *Page 10, Lines 21-22: It is not entirely clear what "heating" the authors are talking about. One can guess that this is the winter time heating of homes, but then the audience should not have to guess, right? Is there any analysis of how realistic the seasonal cycle in domestic emissions is?*

**Answer:** We have now clarified that point by writing: **A seasonal cycle has been applied to the emissions of the domestic sector, to reflect changes in domestic heating as a function of temperature.** Streets et al. (2003), which describes the weighting calculations, write that "There is undoubtedly seasonal dependence of [domestic sector] emissions, though this is not easy to determine." They then base their calculations on reported variations in residential energy use so the weights should reflect real variations.

**Reviewer 1:** *Page 10, line 29-30: It is mentioned that results of "that last region" (Rest of the World-RotW) are obtained by adding Europe, East Asia and RotW. Why are all of these added to produce Rest of the World SRFs?*

**Answer:** The original text was indeed confusing. We have clarified that point by writing: **Emission perturbations involve a 20% decrease of primary and precursor emissions of the given species in one of the following regions: Europe, East Asia, shipping, and Rest of the World (RotW). Results for RotW are not presented directly in this paper: instead, global results are given by adding Europe, East Asia, and RotW together.**

**Reviewer 1:** *Page 11, line 1: Reference needed after "climate policy objectives".*

**Answer:** Rephased to **scientific recommendations to air quality and climate policy (Schmale et al., 2014)** and added reference to Schmale et al., Nature, 2014.

**Reviewer 1:** *Page 11, line 7: Referenced needed after "policy agenda".*

**Answer:** Rephrased to **Because of the specific impact of the shipping sector on air quality (Viana et al., 2014) […]** and added reference to Viana et al., Atmos. Environ., 2014.

**Reviewer 1:** *Page 11, lines 15-17: Is it too late to correct these mistakes? These obviously add to the diversity in the SRFs for shipping.*

**Answer:** Unfortunately it is too late because computing resources supported by the ECLIPSE project (which finished in 2015) have been spent. Note however that, as pointed out in the paper, errors in emissions will not propagate fully to the specific radiative forcing, since that quantify is normalised by emission changes. But those errors affect the baseline, which impacts non-linear radiative forcing, such as aerosol-cloud interactions.

**Reviewer 1:** *Page 11, lines 17-21: Differences in the VOC species considered by the models (because of differences in chemical mechanisms implemented in the model) are a significant source of diversity in the simulated tropospheric ozone (Young et al., 2013). This should be highlighted here.*

**Answer:** Agreed. The section now reads **As discussed in Sect. 3.3, differences in the VOC species included in the models add to SRF diversity.** Reference to Young et al. (2013) is now made in section 3.3 when discussing the impact of different VOC species.

**Reviewer 1:** *Page 11, lines 23-25: Do the models prescribe global mean surface CH4 concentrations or apply a latitudinal variation?*

**Answer:** They prescribe the global mean. This is now specified in the paper.

**Reviewer 1:** *Page 13, 1st paragraph: How do methane lifetimes compare across models? ACCMIP studies have revealed diversity in simulated OH and CH4 lifetimes (Naik et al., 2013; Voulgarakis et al., 2013). Do the models considered here also suffer from this diversity?*

**Answer:** Methane lifetimes are given in Table 7, and a statement in section 3.2 now draws attention to their diversity: **Simulated methane lifetimes vary by a factor 1.6, reproducing the diversity seen in past studies (Voulgarikis et al., 2013)**. Note however that the prescription of surface concentrations and the short (1 year) simulations used in the study should minimise the impact of differences in methane lifetime on specific radiative forcings.

**Reviewer 1:** *Page 13, line 3: "BC lifetime is longer" than what? The statement is not clear.*

**Answer:** We meant to say longer than sulphate lifetime, but comparisons of lifetimes between species have now been removed for the sake of brevity.

**Reviewer 1:** *Page 13, lines 7-10: This is a very broad-brush way of dealing with diversity. Can we not learn anything about what drives diversity in lifetimes from further analysis of model output?*

**Answer:** Not without sensitivity experiments where differences in transport, deposition, etc. can be isolated. Those experiments have not been done within ECLIPSE or, more generally, in past model inter-comparison studies like AeroCom or ACCMIP. There have been single-model studies that looked at processes controlling lifetime (e.g. Kipling et al., doi:10.5194/acp-16-2221-2016, 2016), but to do so in a multi-model context is a sizeable undertaking that falls out of the scope of the present paper.

**Reviewer 1:** *Page 13, lines 18-22: Please qualify these statements with references or point the reader to a particular table that lists the papers that evaluated ECLIPSE models.*

**Answer:** The statement has been rephrased to **Aerosol and ozone distributions simulated by the four models participating in this study have been compared to observations as part of their**

**development cycles (Bellouin et al., 2011; Kirkevåg et al., 2013; O'Connor et al., 2014; Skeie et al., 2011; Zhang et al., 2012), multi-model inter-comparisons (Koffi et al., 2016; Pan et al., 2014; Stevenson et al., 2013; Tsigaridis et al., 2014), and within the ECLIPSE project (Eckhardt et al., 2015; Quennehen et al., 2016; Schulz et al. 2015).**

**Reviewer 1:** *Biases and scaling of specific radiative forcing: Please consider condensing this section. There is no new evaluation presented in the two paragraphs on page 14. The discussion is referring to results from other papers, which can be succinctly summarized in section 3 to explain the results.*

**Answer:** The discussing has been condensed to two paragraphs, which more succinctly summarise comparisons to aerosol and ozone observations.

**Reviewer 1:** *Section 3 - How do the SRFs for O3 and its precursors calculated here compare with estimates from previous studies including - Naik et al. (2005), Fry et al., (2012), Fry et al. (2013). Fry et al. (2014)*

**Answer:** Those studies are indeed very relevant, thank you. We decided to focus on Fry et al. (2012), which is a multi-model study very comparable to those already included in Table 1. The estimates from Fry et al. (2012) have therefore been added to Table 1 and discussed in the introduction: **Regionally for ozone precursor perturbations, Fry et al. (2012) find that South Asia exerts the strongest SRF for NOX and VOC perturbations, while CO perturbations exhibit little regional dependence. Aerosol contributions to net ozone precursor SRFs vary in both sign and magnitude among models, and also regionally.**

Differences between the present study and Fry et al. (2012) are then discussed in more details in Section 4: **Compared to Fry et al. (2012), this study quantifies aerosol responses to ozone precursor perturbations for more aerosol species and RF mechanisms, especially including aci. Those additional components put the aerosol contribution more firmly into negative values for NOX and VOC perturbations, but with increased model diversity. For CO perturbations, Fry et al. (2012), which only accounted for sulphate RFari, found that aerosols contributed a negative SRF. This study finds that that contribution may in fact be positive because nitrate aerosols more than compensate for the sulphate RF.**

**Reviewer 1***: Page 16, lines 21-23, 28-30: I am not sure if I follow the discussion here. On lines 21-22, it is mentioned that the ari is stronger in winter than in summer but then on line it is mentioned that the SO2 is SRF is stronger for summer than winter. This appears contradictory.*

**Answer:** Those statements have been removed for the sake of brevity but for reference, the first statement referred to the contribution of ari to total RF, not RFari itself. RFari is stronger in summer but its contribution to total RF peaks in winter.

**Reviewer 1:** *Page 18, lines 15-17: Please insert a reference here to support this statement that O3 and CH4 RF are affected by BC perturbations via heterogeneous reactions.*

**Answer:** The statement is really that such reactions represent a negligible component of ozone and methane RF. The mechanism is that sulphate and second organic carbon aerosols form a coating onto black carbon particles, altering the chemical characteristics of their surface. So changing the amount of black carbon particles alters the thickness and properties of their coating. Experiments with NorESM1 found a negligible impact on radiative forcing, but those results have not been published. Previous studies, such as Liao et al., *J. Geophys. Res.*, doi:10.1029/2003JD004456, 2004, do not focus on black carbon. For those reasons, the statement has been removed from the revised paper.

**Reviewer 1:** *Page 22, section 3.3: Are secondary organic aerosols (SOA) included in the quantification of OC SRFs?*

**Answer:** Only primary emissions of OC aerosols were perturbed, but this may indirectly affect SOA formation by changing the number of particles on which organic compounds condensate.

**Reviewer 1***: Page 38, lines 19-21: There are several studies in the literature that have studied regional O3 SRFs some of which have been highlighted in my comments above. These need to [be] acknowledged here.*

**Answer:** Agreed. We now discuss Fry et al. (2012): **The regional dependencies of CO perturbations are however weaker than those of NO$_X$ and VOC, as also found by Fry et al. (2012).**

**Reviewer 2:** *Page 5, lines 22-25: since several forcers have negative forcings, it would be better to list the ones with the positive forcings instead of using "All"*

**Answer:** We agree that the original phrasing was awkward and the statement now reads **Black carbon (BC) aerosols, methane, CO and volative organic compounds (VOCs) exert positive SRFs, which lead to a gain in energy for the climate system when emissions are increased. In contrast, sulphate, organic carbon (OC), and nitrate aerosols, and nitrogen oxides (NOX), exert negative SRFs.**

**Reviewer 2***: Page 12, lines 13-19: this definition would not be sufficient for computing RFaci (see Ghan, 2013; http://www.atmos-chem-phys.net/13/9971/2013)*

**Answer:** Steve Ghan's paper refers to calculations that approximate RFaci as the difference in cloud radiative forcing (all-sky minus clear-sky radiative fluxes) between two simulations. We do not do that approximation: our radiative forcings are computed as differences in all-sky radiative fluxes only, so will not be biased by aerosol-radiation interactions happening above clouds. We have however

clarified the methodology by rephrasing the statement to "The method used to achieve this independence involves diagnosing radiative fluxes with and without **the perturbation to** the forcing agent included, with the second set of radiative fluxes used to advance the model into its next time step." So in the case of RFaci, radiative fluxes are never computed with respect to a no-aerosol atmosphere.

**Reviewer 2:** *Page 23, lines 26-27: this kind of comment has to be substantiated.*

**Answer:** To our knowledge there has not been a study dedicated to aerosol-cloud interactions by nitrate aerosols alone, so our statement is in fact an assumption, and has been rewritten to make that point clear: **Diversity of aerosol-cloud interactions for nitrate is assumed to be similar to the 10% obtained in this study for sulphate aerosols.**

**Reviewer 2:** *Similarly, Page 25, lines 16-18*

**Answer:** That statement was difficult to substantiate without knowledge of RFaci in OsloCTM2 for methane perturbations, which we do not have. So the statement has been removed and the paper instead reads: **The OsloCTM2 value is from a simplified calculation, which only represents ari by using distributions of radiative forcing efficiencies instead of the full radiative transfer calculations normally used.**